# Inhibition of delta-secretase improves cognitive functions in mouse models of Alzheimer's disease

Zhentao Zhang[1,2,*], Obiamaka Obianyo[1,*], Elfriede Dall[3,*], Yuhong Du[4], Haian Fu[4], Xia Liu[1], Seong Su Kang[1], Mingke Song[5], Shan-Ping Yu[5], Chiara Cabrele[3], Mario Schubert[3], Xiaoguang Li[6], Jian-Zhi Wang[6,7], Hans Brandstetter[3] & Keqiang Ye[1]

δ-secretase, also known as asparagine endopeptidase (AEP) or legumain, is a lysosomal cysteine protease that cleaves both amyloid precursor protein (APP) and tau, mediating the amyloid-β and tau pathology in Alzheimer's disease (AD). Here we report the therapeutic effect of an orally bioactive and brain permeable δ-secretase inhibitor in mouse models of AD. We performed a high-throughput screen and identified a non-toxic and selective δ-secretase inhibitor, termed compound 11, that specifically blocks δ-secretase but not other related cysteine proteases. Co-crystal structure analysis revealed a dual active site-directed and allosteric inhibition mode of this compound class. Chronic treatment of tau P301S and 5XFAD transgenic mice with this inhibitor reduces tau and APP cleavage, ameliorates synapse loss and augments long-term potentiation, resulting in protection of memory. Therefore, these findings demonstrate that this δ-secretase inhibitor may be an effective clinical therapeutic agent towards AD.

[1] Department of Pathology and Laboratory Medicine, Emory University School of Medicine, Atlanta, Georgia 30322, USA. [2] Department of Neurology, Renmin Hospital of Wuhan University, Wuhan 430060, China. [3] Department of Molecular Biology, University of Salzburg, Salzburg A-5020, Austria. [4] Department of Pharmacology, Emory Chemical Biology Discovery Center, Emory University School of Medicine, Atlanta, Georgia 30322, USA. [5] Department of Anesthesiology Emory University School of Medicine, Atlanta, Georgia 30322, USA. [6] Pathophysiology Department, School of Basic Medicine and the Collaborative Innovation Center for Brain Science, Key Laboratory of Ministry of Education of China for Neurological Disorders, Tongji Medical College, Huazhong University of Science and Technology, Wuhan 430030, China. [7] Co-innovation Center of Neuroregeneration, Nantong 226001, China. * These authors contributed equally to this work. Correspondence and requests for materials should be addressed to J.-Z.W. (email: wangjz@mails.tjmu.edu.cn) or to H.B. (email: Johann.Brandstetter@sbg.ac.at) or to K.Y. (email: kye@emory.edu).

δ-Secretase, also known as asparagine endopeptidase (AEP) or legumain, is a lysosomal cysteine protease that cleaves peptide bonds C-terminally to asparagine residues. δ-secretase is involved in various cellular events, including antigen processing, the cleavage of other lysosomal enzymes, osteoclast formation and normal kidney function[1–5]. Biochemically, the enzyme is highly regulated by its specificity for asparagine residues and pH. Dysregulated δ-secretase activity has been implicated in various diseases, including cancers and neurodegenerative diseases[6–12]. We recently found that δ-secretase is progressively upregulated and activated during ageing in the mouse brain. Moreover, δ-secretase is also elevated and activated in human AD brains compared to normal controls. The active δ-secretase cleaves both amyloid precursor protein (APP) and tau, two major pathogenic players in AD. Processing of APP by δ-secretase facilitates BACE1 to cleave APP, leading to amyloid-β upregulation[13]. On the other hand, active δ-secretase cleaves tau, abolishes its microtubule assembly function, induces tau aggregation and triggers neurodegeneration[14]. Recently, it has also been reported that δ-secretase is responsible for the hyperphosphorylation of tau through its cleavage of SET. After cleavage by δ-secretase, SET fragments inhibit the activity of protein phosphatase 2A (PP2A), which is responsible for 70% of tau phosphatase activity[12]. Thus, these findings support that δ-secretase acts as a crucial mediator in the onset and progression of AD. Inhibition of δ-secretase may be therapeutically useful for treating neurodegenerative diseases including AD.

To date, various inhibitors of δ-secretase have been described. Cysteine proteases are physiologically regulated by interacting proteins, such as cystatins, which form a reversible high-affinity complex with the enzyme[10]. Synthetic inhibitors that have been generated to target δ-secretase, are generally peptide-based inhibitors, which incorporate an asparagine residue for specificity and halomethylketones, Michael acceptors or acyloxymethylketones, to inhibit the catalytic activity of the enzyme[15–17]. Such reactive groups are commonly employed for the inhibition of caspases and cysteine proteases and thus do not offer much selectivity. Asparagine derivatives, outside of the context of a peptide, do not seem to be effective inhibitors of δ-secretase. However, tri- and tetrapeptides seem to be preferred[15]. Despite their favourable qualities, such as high solubility and target specificity, peptides are not generally considered as ideal drug candidates, because they are subject to proteolytic degradation and tend to exhibit low-bioavailability. A DNA vaccine targeting δ-secretase was able to effectively inhibit the enzyme and suppress breast tumour growth and metastasis in mice[18]. However, an effective small molecule inhibitor of δ-secretase has yet to be identified. Novel inhibitors of the enzyme might be useful as powerful therapeutic agents. Moreover, small-molecule inhibitors of δ-secretase could be used to attenuate neuronal death and ultimately, prevent neurodegenerative diseases. Herein, we describe the discovery of small-molecule δ-secretase inhibitors using high-throughput screening and their *in vitro* and *in vivo* characterization. Using high-throughput screen, we identified compound 11 as a potent and specific small molecular inhibitor of δ-secretase. Co-crystal structure analysis revealed that compound 11 interacts with both the active site and allosteric site of δ-secretase. Chronic treatment with compound 11 attenuated tau and APP cleavage in tau P301S and 5XFAD transgenic mouse models, respectively, and ameliorated synaptic dysfunction and cognitive impairments.

## Results

**High-throughput screen for inhibitors of δ-secretase**. To identify small-molecule inhibitors of δ-secretase, we designed a high-throughput screen in conjunction with the Emory Chemical Biology Discovery Center. The screen incorporated mouse kidney lysates to assay a 54,384 compound library. On counter-screening with kidney lysates from δ-secretase knockout mice, 736 hits were confirmed to display $IC_{50}$ values towards the cellular δ-secretase less than or equal to 40 μM. A third screen with purified active δ-secretase found that 46 hits exhibited promising inhibitory activity (Supplementary Fig. 1; Supplementary Table 2). Structural analysis and grouping allowed the compounds to be sorted into eight distinct substructure families. After the most potent compounds from each group were tested with purified active δ-secretase, $IC_{50}$ values for the top 8 candidates were calculated. The specificity of the compounds was also determined using four major cysteine proteases (Fig. 1a,b). Compound BB1 appeared to possess the greatest potency toward δ-secretase, at ~130 nM, and it was about 38-fold more selective for δ-secretase than cathepsin-L, although its ligand efficiency of 0.75 could indicate covalent reaction with δ-secretase[19]. Compound 22 on the other hand, had an $IC_{50}$ greater than 100 μM for all of the tested cysteine proteases. Compounds 11 and 38 were potent inhibitors of δ-secretase, since they displayed $IC_{50}$ values of ~700 and 370 nM, respectively, and they were at least 80-fold more selective for δ-secretase than caspase-3 or caspase-8. Although compound 64 exhibited the highest $IC_{50}$ at 2.37 μM, it was still a low-micromolar inhibitor and it was over 40-fold more selective for δ-secretase as compared to cathepsin-S, cathepsin-L and caspase-8 and about 6-fold more selective as compared to caspase-3 (Fig. 1b). In an attempt to assess the activity of the compounds in intact cells, the compounds were incubated with human B lymphoblastoid Pala cells, which are rich in endogenous δ-secretase and have been used previously for δ-secretase inhibitor analysis in living cells[16]. Most of the compounds were able to inhibit the enzyme with $IC_{50}$ values in the sub-micromolar range. However compounds BB1 and 10 exhibited slightly larger $IC_{50}$ values. Nonetheless, all of the compounds seem to be cell permeable (Fig. 1c; Supplementary Fig. 2a–g). Since the δ-secretase activity assays are fluorescence-based, and compound 11 has been reported as active in several other fluorescence-based assays listed in PubChem, we tested whether compound 11 absorbs the fluorescence of AMC. We found that compound 11 up to 7 μM (10 times of IC50) did not affect the fluorescent reading of AMC, thus precluding the potential for fluorescence interference (Supplementary Fig. 2h). We chose to determine the absorption, distribution, metabolism, excretion, and toxicity (ADMET) characteristics of the compounds with $IC_{50}$ values < 1 μM.

**In vitro ADME profiles and toxicity of the lead compounds**. To determine which lead compounds possess the drug-like characteristics and should thus be a focus of subsequent analyses, we performed *in vitro* ADMET assays of the compounds with the most promising structural backbones. First, a Caco-2 monolayer permeability screen was performed to assess the absorption characteristics of the compounds. Compounds 11 and 31 were found to be highly permeable and should therefore be well physiologically absorbed (Supplementary Table 3). However, in the BBB-PAMPA permeability assay, only compound 11 was detected at high-levels and was thus considered able to cross the blood-brain barrier (Supplementary Table 3). The human liver microsomal stability screen demonstrated that following 30 min of incubation, 76% of compound 11 and 88% of compound 38 remained in human liver microsomes, while the other compounds were relatively metabolically unstable (Supplementary Table 4). According to cytochrome P450 (CYP) enzymes inhibition screening, compound 31 was able to inhibit CYP2C9 at 69.2% and CYP2C19 at 55.7%, while compound 64 inhibited

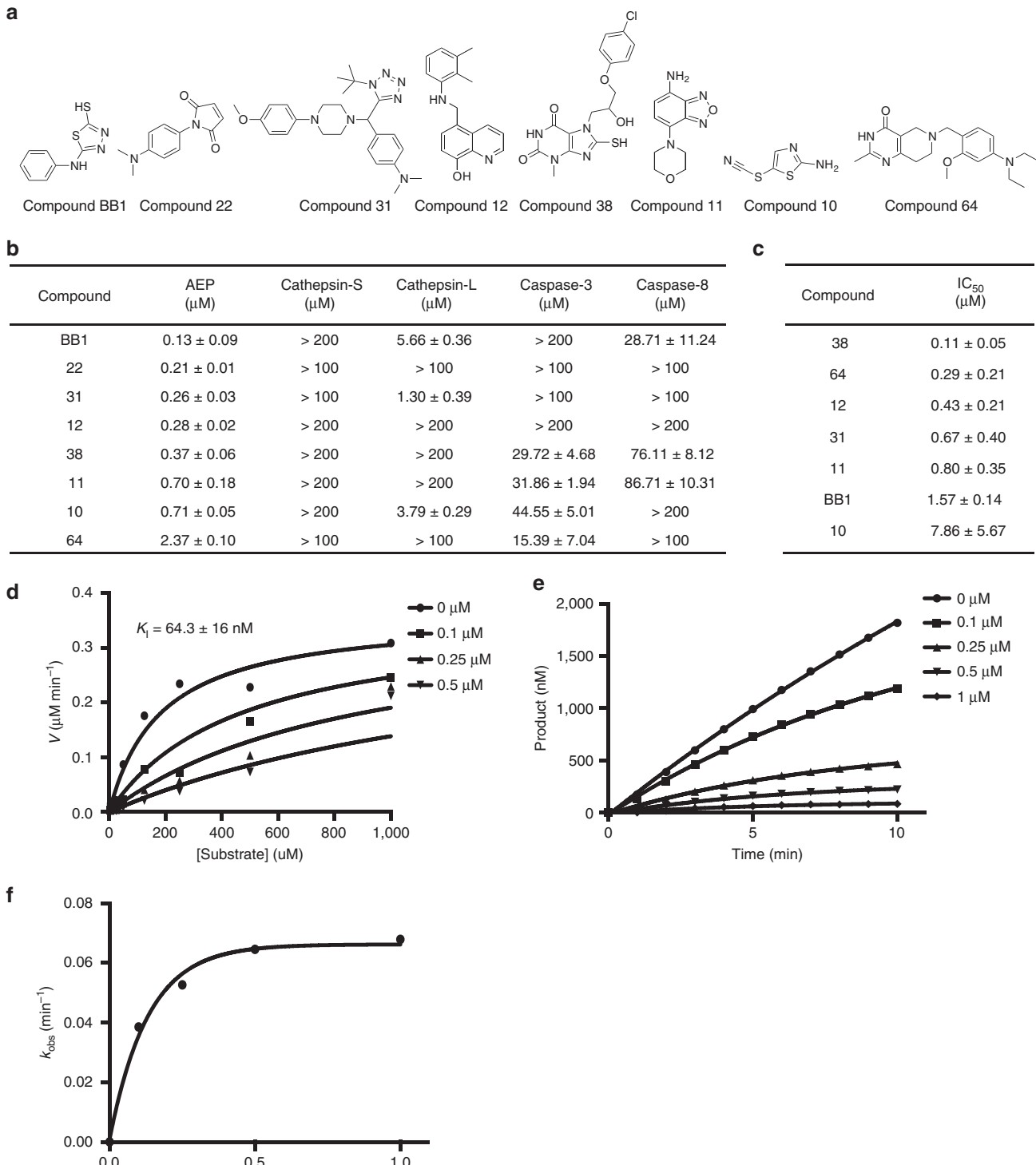

**Figure 1 | Characterization of the compounds. (a)** Chemical structure of the top 8 candidates. **(b)** IC$_{50}$ values of the compounds towards δ-secretase, cathepsin-S, cathepsin-L, caspase-3 and caspase-8 (mean ± s.d.). **(c)** IC$_{50}$ values of the compounds towards δ-secretase in Pala cells (mean ± s.d.). **(d)** Steady-state kinetic parameters were determined by varying substrate, Z-AAN-AMC, at fixed concentrations of compound 11. $K_I$ value was determined by global fits to the competitive inhibition equation. **(e)** Time course inactivation assays were used to generate progress curves, depicting product formation as a function of time. Pseudo first-order rate constants were obtained at each concentration of compound 11. **(f)** Re-plot of the pseudo first-order rate constants, $k_{obs}$, versus the concentration of compound 11.

CYP2D6 at 58.3% at 10 μM of concentration, suggesting that these two compounds might be capable of producing potential drug-drug interactions (Supplementary Table 5). Taken together, these ADME results demonstrate that compound 11 possesses the most promising characteristics as the lead compound for treating CNS diseases on oral administration.

In an effort to characterize the toxicity of the compounds, we performed an MTT assay using human hepatocellular carcinoma,

HepG2 cells and primary neurons to monitor the cell viability. In HepG2 cells, only compound 22, the maleimide-containing derivative, revealed a similar toxicity to the positive control etoposide, which is a topoisomerase inhibitor and known to induce double strand breaks (Supplementary Fig. 3a). We also determined the cytotoxicity of the compounds in primary neuronal cultures, using an LDH assay to measure cytolysis. None of the compounds seems to be toxic to the neurons (Supplementary Fig. 3b). Similarly, none of the compounds exhibited the cytotoxicity in human embryonic kidney 293 (HEK293) cells and B lymphoblastoid Pala cells (Supplementary Fig. 3c–f). The carcinogenicity of a compound is directly proportional to its induction of micronuclei[20]. To assess whether the compounds possess any carcinogenicity, we performed a COMET assay and a micronucleus assay in HepG2 cells. Benzo(a)pyrene (B(a)p), which generates measurable DNA nicks within these assays was included as a positive control. Following treatment for 24 h with 50 μM compound, compound 22 was the only test compound to show genotoxicity. Thus, it was excluded from further analyses (Supplementary Fig. 3g,h).

**Inhibitor characterization**. To gain insight into the mechanism utilized by the inhibitors for abrogation of δ-secretase activity, their reversibility in the presence of free thiols was determined. As previously reported, the addition of a strong reducing agent, such as DTT or a weaker reducing agent, L-cysteine, to an inhibited reaction can be used to out-compete the inhibitory agent and restore catalytic activity to an enzyme with an active-site thiol residue[21]. Here, we used a similar approach by adding a reducing agent, either DTT or L-cysteine, to a reaction, in which δ-secretase had been incubated with a specified inhibitor, in an attempt to reverse the effects of the inhibitor. None of the inhibitors could be completely out-competed. In the presence of the thiol-containing compounds, BB1 and 38 and compound 10, which contain a thiocyanate moiety, δ-secretase regained a substantial amount of activity on addition of reducing agents, indicating that the reducing agents were able to reduce the active-site cysteine of the enzyme and thus increase the effective concentration of active enzyme (Supplementary Fig. 4). For compounds BB1 and 38, this increase in enzymatic activity may be due to the reduction of a disulfide linkage between the inhibitor and the enzyme, since these compounds contain thiols. Compound 10 contains a thiocyanate warhead, and most likely undergoes nucleophilic attack by the enzyme's active-site thiolate to form a thioimidate enzyme-inhibitor complex. Under acidic conditions, this complex is readily reducible by either a strong reducing agent, such as DTT, or the weaker L-cysteine[22]. Thus, it is possible that compounds BB1, 10 and 38 may form covalent bonds with the active-site cysteine of δ-secretase and competitively inhibit its activity.

The identified compounds contain structural moieties with potential thiol reactivity, similarly to pan-assay interference (PAINS) compounds, which might lead to unspecific off-target interferences[23–26]. Among 8 positive hits skeletal backbone structures, most of them (BB1, #10, #12, #22 and #64) contain well-characterized PAINS or potential thiol-reactive chemotype. The rest of hits compound #31 and #38 display some of the favourable in vitro ADMET properties including caco-2 absorption, metabolic stability and so on, but they barely penetrate the BBB (Supplementary Table 3). Hence, we focused on the thiol free compound 11 for activity studies. Although, this compound contains some susceptible PAINS motifs (benzofurazan), the SAR assays (Supplementary Fig. 5a–e) demonstrate that this compound might not act as PAINS but

rather specifically blocks δ-secretase protease activity. A nitro group in place of the primary amine (11d15) fully inactivated compound 11. Amide conjugation on the primary amine (derivative #1-14 but not #7) weakened compound 11's activity, whereas alkylation (derivative #16-23) somehow escalated its effect. On the other hand, replacing the morpholino group with pyrrolidine (#24) or piperidine (#25) greatly reduced compound 11's inhibitory activity. Noticeably, elimination of the furazan group in #26 substantially diminished compound 11's activity, but replacing it with another morpholino group in #27 (compound 11b) augmented its activity (Supplementary Fig. 5d,e).

To determine the mode of inhibition of compound 11, steady-state kinetic parameters were measured in the presence of increasing concentrations of the compound (Fig. 1d). The inhibitor constant, $K_I$, is the concentration of inhibitor that is required to produce half-maximal inhibition and is a measurement of an inhibitor's potency[27]; the $K_I$ for compound 11 is $64 \pm 16$ nM. To further characterize its mode of inhibition, progress curves were measured at increasing concentrations of compound 11. The resulting curvilinear plots indicate that compound 11 inhibits δ-secretase in a concentration- and time-dependent manner (Fig. 1e). In addition, plotting the pseudo-first-order rate constants of inhibition, $k_{obs}$, which were determined from the progress curves, yielded a hyperbolic curve, consistent with a two-step mechanism of inactivation, which occurred on two different time scales[28,29] (Fig. 1f). The rate of inactivation, $k_{inact}$, was found to be $0.075 \pm 0.002$ (mean ± s.e.m.) min$^{-1}$, thus the apparent second-order rate constant, $k_{inact}/K_I$ is $1.2 \times 10^6$ min$^{-1} \cdot$ M$^{-1}$ suggesting that compound 11 inactivates δ-secretase very quickly and potently.

**Characterization of δ-secretase inhibition by compound 11**. The kinetic analysis showed a bi-phasic inhibition mode of compound 11, that is a fast-loose slow–tight interaction with δ-secretase. Such bi-phasic inhibition can be rationalized by conformational rearrangements within the protein–inhibitor complex and are often observed with allosteric inhibition[30]. To directly assess the inhibition mode, we immobilized compound 11 and the structurally related compound 11b and tested its binding to different human δ-secretase variants by the surface acoustic wave (SAW) method. The tested δ-secretase variants included active site-blocked forms, namely prolegumain (proC189S), human cystatin E-inhibited (C189S-δ-secretase + hCE) and covalently inhibited δ-secretase (δ-secretase-YVADcmk), as well as δ-secretase forms with freely accessible active site (δ-secretase; C189S-δ-secretase). Importantly, all active site blocked δ-secretase forms exhibited significant binding, demonstrating that compound 11 does not only bind to δ-secretase's active site but rather targets a regulatory exosite of δ-secretase (Fig. 2a). Cystatin E (hCE) and the cysteine protease papain served as a negative control to exclude non-specific binding by compound 11 and 11b. A more detailed analysis of the binding sensogram revealed that the pre-occupation of the δ-secretase active site by a substrate analogue retards both the association and dissociation process, as reflected by markedly shallower binding and dissociation curves. Quite likely, this influence is reciprocal in nature and is mediated by a conformational cross talk between inhibitor binding to the regulatory exosite and the active site, which is stabilized by active site ligands.

To decipher its structural mechanism, we determined the crystal structure of δ-secretase in complex with compound 11. To deduce the expected allosteric inhibition mode, we used δ-secretase that was covalently labelled by the chloromethylketone

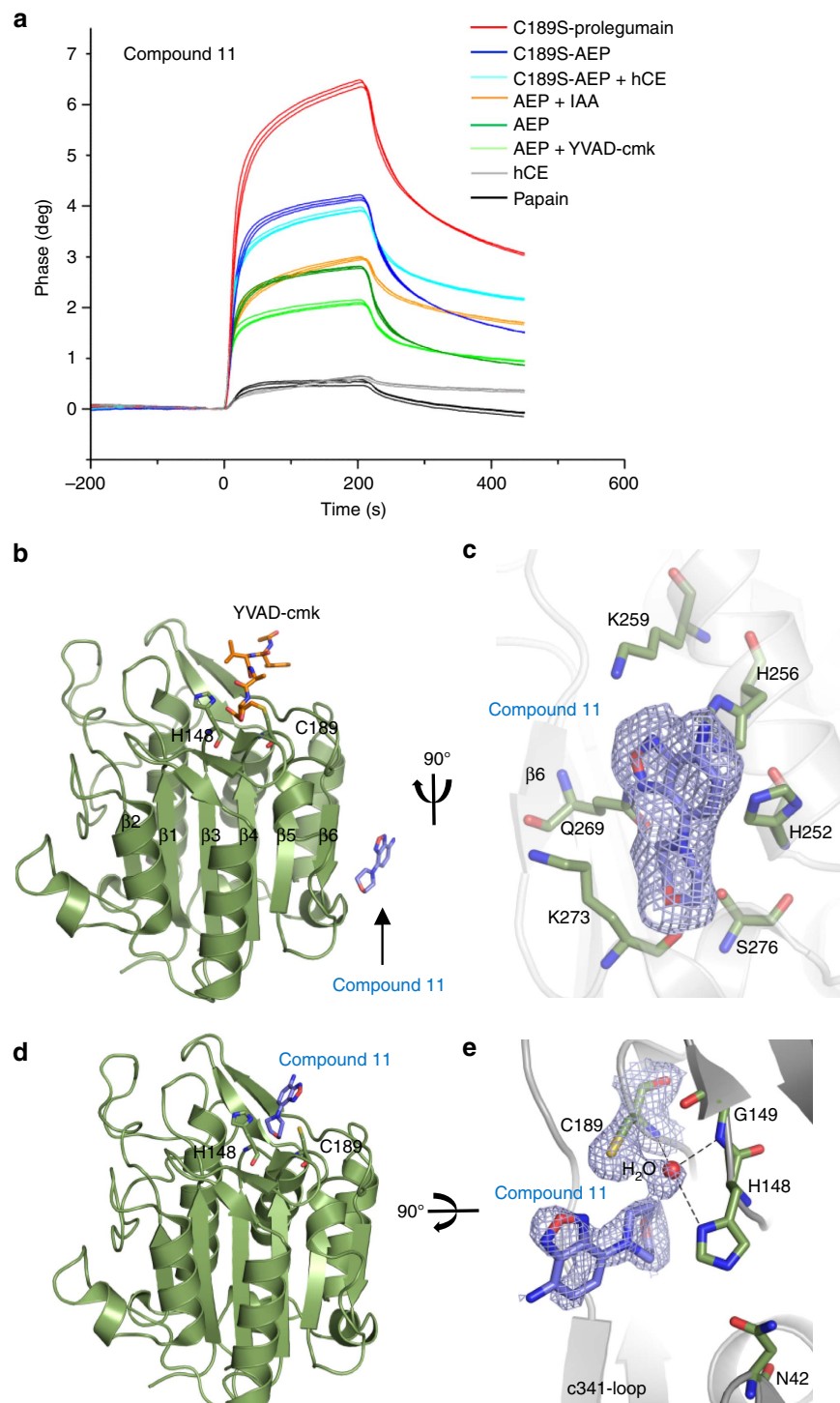

**Figure 2 | Binding of compound 11 to δ-secretase.** (**a**) Both active site liganded and free δ-secretase bind to compound 11 which was immobilized on a sam5BLUE biosensor chip. Binding of C189S-prolegumain (red curves), *in trans* activated C189S-legumain only (dark blue curves) and complexed to cystatin E (light blue curves), δ-secretase (dark green curves) and δ-secretase covalently inhibited with YVAD-chloromethyleketone (δ-secretase + YVAD-cmk, light green curves), was tested at pH 5.0. Cystatin E (grey curves) and papain (black curves) served as a control to test for unspecific binding to the chip. (**b**) Compound 11 binds to δ-secretase at a site close to the dimer-interaction site in caspases. Active site inhibited δ-secretase was crystallized with compound 11. The active site is labelled with the covalent YVAD-cmk inhibitor and shown in orange sticks, the catalytic Cys189 and His148 in green sticks and compound 11 in blue sticks. (**c**) Zoom-in view on the compound 11 binding site. (**d**) Compound 11 binds to the δ-secretase active site. (**e**) Zoom-in view on the active site. The morpholino group binds into δ-secretase's S1-pocket. Furthermore, the oxyanion-pocket, formed by Cys189, Gly149 and His148, is also occupied in the structure. The electron density (2F$_{obs}$–F$_{calc}$) defining compound 11 is contoured at 1σ over the mean. Interacting residues on δ-secretase are shown as green sticks.

peptide YVADcmk. We identified the allosteric inhibition site flanking the β6 strand, located ∼30 Å away from the active site. These findings are fully consistent with the SAW measurements (Fig. 2b,c, Supplementary Fig. 6a and Supplementary Table 1). The protein–inhibitor interaction is of mixed character. The benzofurazan is involved in π–π type interactions with Gln269 (strand β6), as well as in cation-π type interactions by His252, His256 and Lys259, which are harboured by the helix α5 (ref. 31). The morpholine ring forms mostly hydrophobic contacts with the aliphatic part of the side chain Lys273. Additional contacts are mediated by a solvent molecule that crosslinks the morpholine nitrogen to the side chains of His252, Gln269 and Ser276, the latter being stabilized by the carbonyl of Lys273. A sulfate molecule is further interacting with the Nε1 atom of the His252 side chain; all these contacts are strictly conserved between human and mouse (Supplementary Fig. 6b–e).

In a next step, we determined a complex structure of compound 11 with uninhibited δ-secretase. We found the allosteric site nearly identically as in the covalently blocked δ-secretase; additionally, however, compound 11 was bound to the active site. The morpholino-ring binds into the S1 pocket, mimicking the interaction of an Asn side chain of a peptidic substrate, whereas the benzofurazan bound to the S2 pocket (Fig. 2d,e; Supplementary Fig. 7b). Importantly, the benzofurazan was unmodified and did not covalently bind to the active site Cys189. However, non-specific interactions of benzofurazan scaffolds have been reported[24]. Therefore, we have exploited the available crystal structure information and attempted to further develop compound 11 to eliminate the benzofurazan while maintaining or improving the active site and allosteric site interactions. A bis-morpholine-aniline ring system (compound 11b) is able to not only maintain the S1 site interaction but also to mimic the canonical P2–S2 and even P3–S3 interaction at the active site (Supplementary Fig. 7). Furthermore, the interaction at the allosteric site was maintained, with slightly improved electrostatic binding with Lys273, consistent with an ∼2-fold increased binding affinity (Supplementary Fig. 7a). The purity and identity of pro-δ-secretase and activated δ-secretase are shown in Supplementary Fig. 7g. Intriguingly, the identified allosteric binding site in δ-secretase is closely related to allosteric regulatory exosites that have been reported in caspases adjacent to the β6 strand where it mediates dimerization and zymogen-protease conversion (Supplementary Fig. 6d). It is, therefore, plausible that the β6 strand similarly serves as a regulation element that may selectively modulate δ-secretase's protease and ligase activity.

To confirm the purity and stability of compound 11 and 11b, we obtained them from customer-synthesis companies, and confirmed their chemical identity and purity using HPLC, MS and NMR spectroscopy (Supplementary Fig. 8; Supplementary Tables 6,7). Moreover, we validated by NMR spectroscopy that both compounds 11 and 11b were stable in the presence or absence of cysteamine (Supplementary Fig. 9a,b). We also tested their redox potential, and found that none of them produced $H_2O_2$ in the presence or absence of DTT (Supplementary Fig. 9c,d). Notably, compound 11 bound δ-secretase reversibly, but it did not react with the thiol-containing β-mercaptoethanol (Supplementary Fig. 10), further supporting that it does not act as a PAINS.

**Specific inhibitory efficacy in a cellular model.** Depriving cells of oxygen and glucose is a well-established *in vitro* model of hypoxic-ischemic injury[32]. We determined the efficacy of δ-secretase inhibitors in a cellular model of oxygen-glucose deprivation (OGD) in primary cultured neurons. OGD induced

acidosis in cultured neurons (Fig. 3a). As a result, the δ-secretase activity was doubled (Fig. 3b, 'Normoxia' compared to 'OGD'). Compound 11 and its active derivative 11b suppressed δ-secretase activity in a dose-dependent manner, while the inactive derivative with an electron-withdrawing nitro-group on the benzofurazan ring, a standard PAINS, 11d15 did not affect δ-secretase activity (Fig. 3b, Supplementary Fig. 5b,c). As expected, OGD treatment induced increased LDH activity in the medium, suggesting massive neuronal injury. Compound 11 and 11b attenuated neuronal injury induced by OGD, whereas the inactive derivative 11d15 did not show any protective effect (Fig. 3c). We also tested the effect of OGD on neurons prepared from δ-secretase knockout mice. Deletion of δ-secretase gene protected neurons from OGD. Compound 11 and 11b did not further protect δ-secretase knockout neurons from OGD, indicating that their effects are mediated via inhibiting δ-secretase (Fig. 3c). Since compound 11 has been reported to inhibit GAPDH activity at high-concentrations[33], we tested the specificity of compound 11 by performing a GAPDH activity assay. Although compound 11 evidently antagonized δ-secretase activity at 2 μM concentration, it failed to inhibit GAPDH, although it displayed some inhibitory effect at 50 μM concentration. By contrast, the well-recognized GAPDH inhibitor koningic acid (KA) robustly blocked GAPDH at both concentrations (Fig. 3d). Nonetheless, neither compound 11d15 nor KA showed any neuroprotective effect on OGD-induced cell death (Fig. 3c). Hence, compound 11 and its active derivative 11b specifically target δ-secretase and exert neuroprotective actions.

We recently reported that both tau and APP are substrates of δ-secretase[13,14]. To monitor the effect of compound 11 on the cleavage of tau and APP *in vitro*, we incubated kidney lysates with different concentrations of compound 11, followed by incubation with GST-tau or GFP-APP. As the concentrations of compound 11 gradually increased, tau and APP fragmentation was progressively inhibited, indicating that compound 11 effectively inhibited the cleavage of tau and APP by δ-secretase *in vitro*. In contrast, compound 11d15 or KA did not affect the cleavage of tau and APP (Fig. 3e–h). We further explored the effect of compound 11 on δ-secretase-mediated truncation of tau on OGD treatment using antibodies that specifically recognize δ-secretase-cleaved tau fragment (anti-tau N368)[13,14]. In neurons overexpressing human tau, we found that human tau was readily cleaved by δ-secretase after OGD treatment, which was inhibited by compound 11 in a dose-dependent manner (Fig. 3i). Thus, compound 11 acts as a specific δ-secretase inhibitor, not a non-specific PAINS, to selectively block tau and APP fragmentation by δ-secretase.

**Compound 11 ameliorates tau cleavage in tau P301S mice.** Tau P301S transgenic mice express human tau bearing P301S mutation, and develop filamentous tau lesions at 6 months of age[34]. Previously we found that δ-secretase is activated in tau P301S transgenic mice brain, cleaves tau after N368 and promotes tau aggregation[14]. To test whether oral administration of compound 11 can attenuate the cleavage of tau *in vivo*, we consecutively treated tau P301S mice and non-transgenic control mice with compound 11 or vehicle once daily for 3 months, beginning at 2 months of age. Oral administration of compound 11 significantly inhibited the activity of δ-secretase in both wild-type and tau P301S mice brain (Fig. 4a). Compound 11 did not change the expression level of full-length human and mouse tau, but decreased the cleavage of both endogenous mouse tau and transgenic human tau by δ-secretase after N368, as demonstrated by Western blot using the tau N368 antibody and HT7 antibody. The phosphorylation, aggregation and

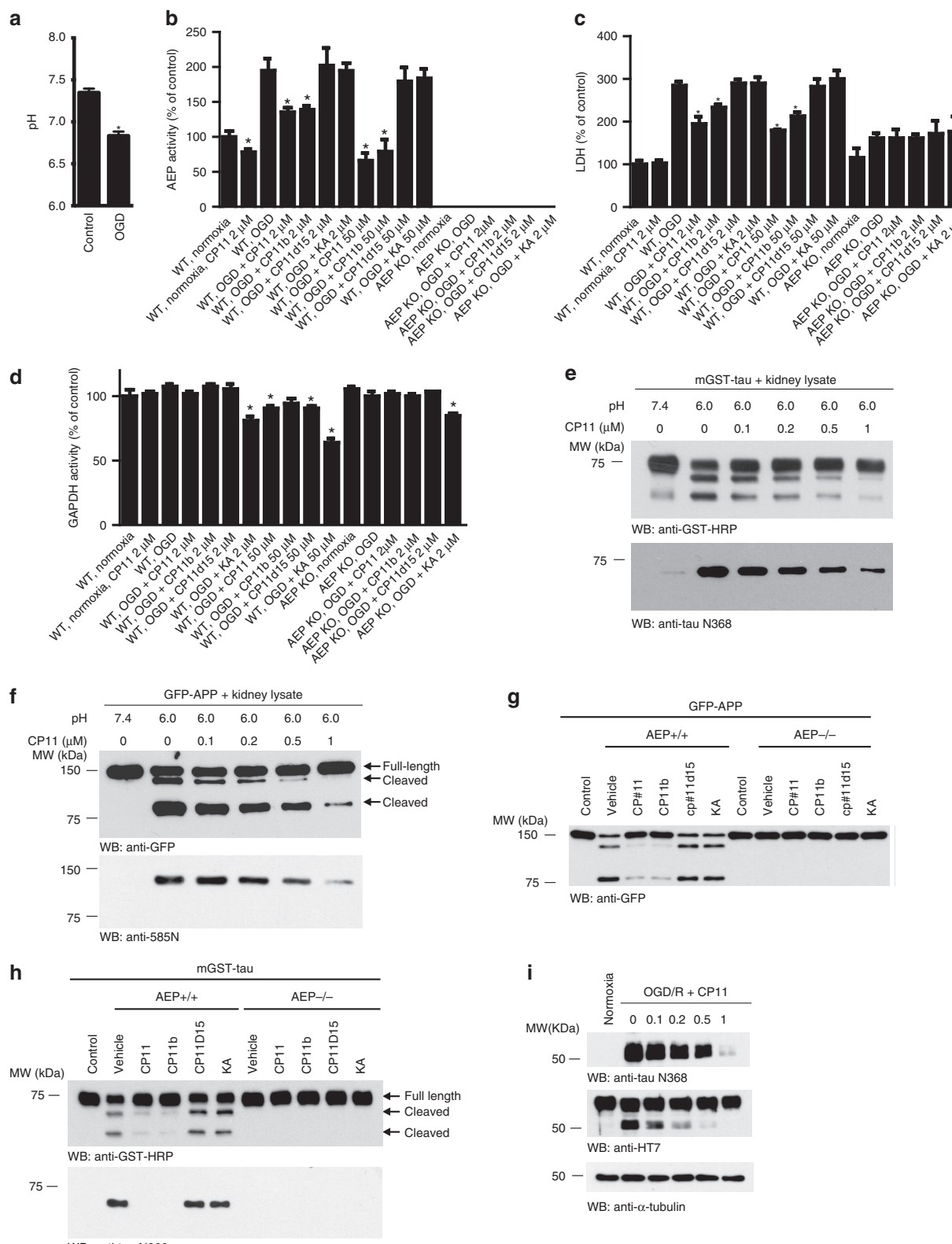

**Figure 3 | Compound 11 inhibits δ-secretase activity *in vitro*.** (**a**) pH values of neuronal lysates before and after OGD treatment (mean ± s.e.m. of three independent experiments). (**b**) δ-secretase activity in primary neurons treated with compound 11, 11b, 11d15 or KA (koningic acid) (mean ± s.e.m. of three independent experiments). (**c**) LDH activity in the medium of primary neurons treated with compound 11, 11b, 11d15 or KA (mean ± s.e.m. of 3 independent experiments). (**d**) GAPDH activity in primary neurons treated with compound 11, 11b, 11d15 or KA (mean ± s.e.m. of three independent experiments). (**e**) Inhibition of GST-tau cleavage by δ-secretase *in vitro*. Kidney lysates were first incubated with different concentrations of compound 11 and then incubated with GST-tau (isoform 4). The cleavage of tau was analysed by Western blot. (**f**) Inhibition of GFP-APP cleavage by δ-secretase *in vitro*. (**g,h**) Effect of compound 11, 11b and 11d15 on the cleavage of GFP-APP and GST-tau by kidney lysates from wild-type ($^{+/+}$) and δ-secretase knockout ($^{-/-}$) mice. (**i**) Inhibition of OGD-induced tau cleavage in neurons overexpressing human tau. *$P < 0.01$. Data were analysed using *t*-test in **a**, and one-way ANOVA followed by post hoc comparison in **b–d**.

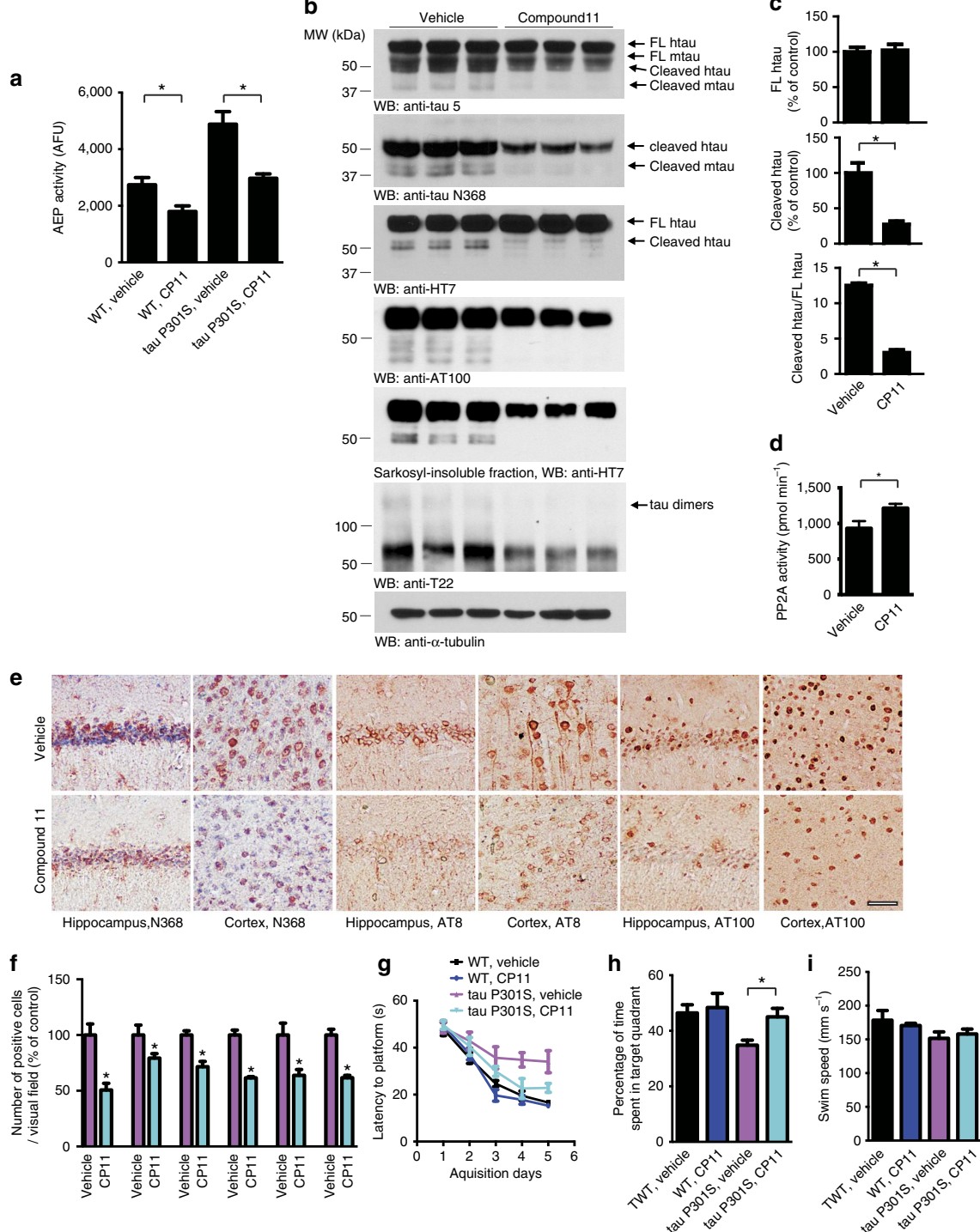

**Figure 4 | Compound 11 inhibits δ-secretase activity and restores cognitive deficits in tau P301S mouse model. (a)** δ-secretase activity assay. Non-transgenic control mice or tau P301S mice were treated with compound 11 or vehicle at 10 mg kg$^{-1}$ d$^{-1}$ for 3 months. Compound 11 significantly decreased the activity of δ-secretase in the brain (mean ± s.e.m.; $n = 4$ mice per group; *$P < 0.01$, one-way ANOVA). **(b)** Western blot showing the processing of tau by δ-secretase. The human tau bands were detected with HT7 antibody, and the rest bands were mouse tau. The phosphorylation of tau was detected using AT100 antibody. The oligomerization of tau was detected using T22 antibody. The aggregated human tau was detected in the Sarkosyl-insoluble fraction. Compound 11 attenuated tau cleavage, phosphorylation, oligomerization and deposition. **(c)** Quantification of cleaved tau and full-length tau in vehicle- and compound 11- treated mice (mean ± s.e.m.; $n = 3$ mice per group. *$P < 0.01$, t-test). **(d)** PP2A activity in vehicle- and compound 11- treated mice (mean ± s.e.m.; $n = 3$ mice per group. *$P < 0.01$, t-test). **(e)** Immunohistochemistry showing the presence of δ-secretase-cleaved tau fragments and AT8-, AT100-positive neurons. Scale bar, 50 μm. **(f)** Quantitative analysis of positive neurons in e (mean ± s.e.m.; $n = 6$; *$P < 0.05$, t-test). **(g)** Morris water maze analysis as latency to platform for mice treated with compound 11 or vehicle (mean ± s.e.m.; $n = 10$ mice per group; *$P < 0.05$, multilevel mixed-effects linear regression model). **(h)** The percentage of time spent in the target quadrant in the probe trail (mean ± s.e.m.; $n = 10$ mice per group; *$P < 0.05$, one-way ANOVA). **(i)** Swim speed of the mice treated with vehicle and compound 11 (mean ± s.e.m.; $n = 10$ mice per group; one-way ANOVA).

oligomerization of tau were attenuated by compound 11 (Fig. 4b,c). Compound 11 treatment also increased PP2A activity, indicating its inhibitor SET was blocked by compound 11 (Fig. 4d). Immunohistochemistry analysis found that the percentage of neurons positive of tau N368 was decreased in both the hippocampus and cortex of compound 11-treated tau P301S mice. Furthermore, the percentage of AT8- and AT100-positive neurons in the hippocampus and cortex was decreased in compound 11-treated tau P301S mice compared to vehicle-treated mice (Fig. 4e,f). These results suggest that compound 11 attenuates the truncation and phosphorylation of tau in tau P301S mice.

*In vivo* pharmacokinetic studies demonstrated that compound 11 oral bioavailability was ∼69.05% with half-life $T_{1/2}$ of 2.31 h (Supplementary Table 8; Supplementary Fig. 11a), supporting that compound 11 is readily absorbed and relatively stable in circulation system on oral administration. After chronic dosing in tau P301S mice, we also detected compound 11 in the serum and brain tissues (Supplementary Table 8), supporting that compound 11 penetrates the BBB and gets access to its cellular target in the CNS.

To investigate whether chronic oral administration of compound 11 can reverse the cognitive deficits of tau P301S mice, we assessed spatial learning and memory using Morris water maze test. During the five acquisition days, the mice treated with compound 11 showed decreased latency to platform when compared with the vehicle-treated mice, indicating improved spatial learning (Fig. 4g). On the probe trial, tau P301S mouse showed less time in the target quadrant than the non-transgenic control mice, indicating impaired spatial memory recall. Notably, the behavioural deficits of tau P301S mice were ameliorated by compound 11 (Fig. 4h). The swim speed was not affected by compound 11 (Fig. 4i). These results indicate that chronic treatment with compound 11 ameliorates memory deficits in the tau P301S mouse model.

**Compound 11 restores synaptic functions in tau P301S mice.** Synaptic dysfunction is an early feature of AD and is believed to be the basis of cognitive impairment[35]. Tau P301S mice show synaptic loss and impaired synaptic function at 3 months of age[34]. To investigate the effect of compound 11 on the synaptic dysfunction, we first assessed the synaptic density in the CA1 area by electron microscopy. Compound 11 significantly increased the density of synapses (Fig. 5a,b). Then we assessed the density of dendritic spines along individual dendrites of pyramidal neurons by Golgi stain. Again, compound 11 increased the density of spines (Fig. 5c,d). Electrophysiological analysis found that the paired-pulse ratios were decreased in tau P301S mice. This electrophysiological impairment was significantly alleviated by compound 11 treatment (Fig. 5e). Long-term potentiation (LTP) is a measure of synaptic plasticity that underlies learning and memory. Compound 11 treatment significantly elevated the LTP amplitude in tau P301S mice (Fig. 5f). These results indicate that compound 11 ameliorates the impaired synaptic function in tau P301S mice, and thus alleviates the deficit in synaptic plasticity (LTP). To further investigate whether compound 11 exerts the beneficial effect via prevention of tau cleavage, we injected AAVs encoding δ-secretase-derived tau fragment 1–368 into the mouse hippocampus, and treated the mice with compound 11 or vehicle. Expression of tau 1–368 fragment caused hyperphosphorylation of tau and cognitive dysfunction. Compound 11 did not affect the phosphorylation of tau, the cognitive function, or the synaptic function of the mice. These results indicate that compound 11 exhibits its therapeutic effect via inhibition of tau cleavage by δ-secretase (Supplementary Fig. 12a–h). Moreover, we found

that compound 11 attenuated OGD-induced cell toxicity in cultured neurons from 5XFAD mice or tau P301S mice, but not in neurons from 5XFAD/AEP$^{-/-}$ mice or tau P301S/AEP$^{-/-}$ mice, further confirming the specificity of compound 11 on δ-secretase (Supplementary Fig. 12i,j).

The chronic toxicity of compound 11 was assessed after the tau P301S mice received the drug at a dose of 10 mg kg$^{-1}$ d$^{-1}$ over a 3-month period via oral gavage. Over the 3 months treatment, no significant change in body weight was observed (Supplementary Fig. 13a). The weight of the major organs including heart, liver, spleen, brain and kidney were not changed by compound 11 (Supplementary Fig. 13b). In addition, hematoxylin and eosin staining of the major organs demonstrated that there were no major differences between vehicle-treated and drug-treated animals (Supplementary Fig. 13c). Complete blood count analysis, renal function tests and liver function tests were also performed on the mice treated with compound 11 and all parameters were comparable with the vehicle-treated mice (Supplementary Table 9). Altogether, these data demonstrate that chronic treatment of compound 11 does not exhibit systemic toxicity in mice at the dosage of 10 mg kg$^{-1}$ d$^{-1}$ for 3 months.

**Compound 11 ameliorates Aβ deposition in 5XFAD mice.** Our most recent data showed that active δ-secretase cleaves APP, facilitates β-secretase-mediated processing of APP and promotes Aβ production[13]. Accordingly, we investigated the effect of compound 11 on 5XFAD mice by treating 2-month-old 5XFAD mice with compound 11 or vehicle once daily for 3 months. 5XFAD mice co-express a total of five mutations associated with familial AD, and develop cerebral amyloid plaques at an early age[36]. Orally administered compound 11 significantly inhibited the activity of δ-secretase in mice brain (Fig. 6a). Compound 11 also decreased the cleavage of APP by δ-secretase at N373 and N585, as demonstrated by western blot using antibodies that specifically recognize δ-secretase-generated APP fragments (Fig. 6b). The expression of C99, the C-terminal fragment of APP generated by β-site cleavage, was decreased after compound 11 treatment, but the expression of α-site cleavage product C83 was not altered by compound 11, indicating compound 11 can attenuate the β-secretase-mediated processing of APP. Compound 11 treatment did not change the protein levels of total APP (Fig. 6b). Thioflavin S staining found that vehicle-treated 5XFAD mice displayed significant plaque deposition in the hippocampus and frontal cortex, which was attenuated by compound 11 (Fig. 6c,d). This result was confirmed by immunohistochemistry staining of amyloid plaque using an anti-Aβ antibody (Supplementary Fig. 14a,b). Furthermore, compound 11 significantly decreased the concentrations of Aβ 1–40 and Aβ 1–42 in the brain lysates (Fig. 6e,f). Next, we treated 5XFAD mice with 2, 5 or 10 mg kg$^{-1}$ d$^{-1}$ of compound 11 for 1.5 and 3 months, respectively, and found that compound 11 attenuated Aβ deposition in a time- and dose-dependent manner (Supplementary Fig. 15a–h). Moreover, single injection of compound 11 also time- and dose-dependently decreased the concentrations of Aβ in both plasma and brain of 5XFAD mice. The concentrations of Aβ in the brain were reversely correlated with the concentrations of compound 11 in the brain (Supplementary Fig. 11b–g). Hence, these results support that compound 11 inhibits the activity of δ-secretase in the brain, and the inhibition of δ-secretase activity decreases the β-site cleavage of APP and the production of Aβ.

The effect of compound 11 on the spatial memory of 5XFAD mice was assessed using Morris water maze test. During acquisition, the mice treated with compound 11 showed decreased distance to platform when compared with the

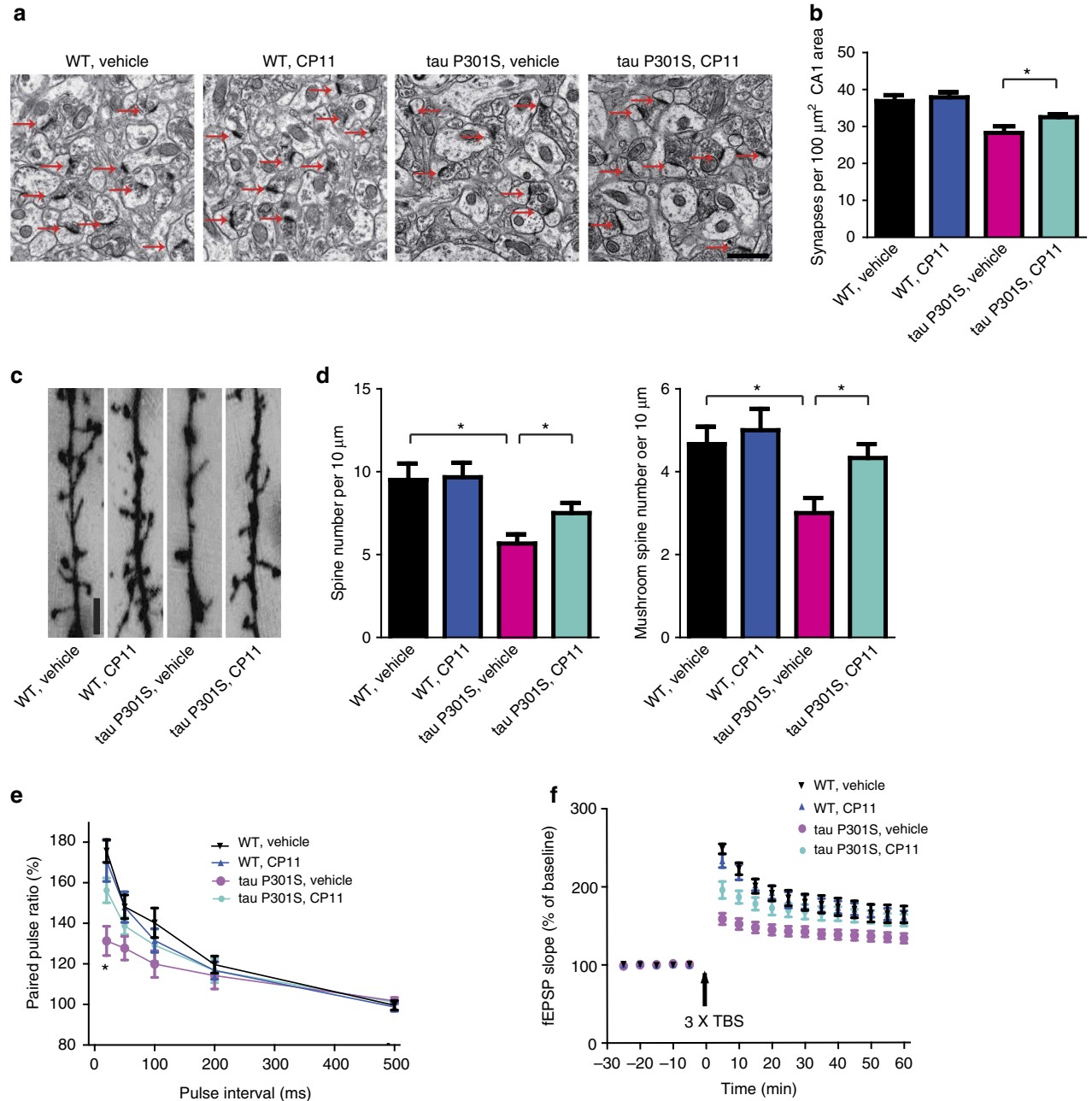

**Figure 5 | Compound 11 ameliorates synaptic loss and restores synaptic dysfunction in tau P301S mice.** (**a**) Representative electron microscopy of the synaptic structures. Arrows indicate the synapses. Scale bar, 1 μm. (**b**) Quantitative analysis of the synaptic density in vehicle- and compound 11-treated control and tau P301S mice. Data are shown as mean ± s.e.m. ($n = 4$ mice per group; $*P < 0.05$, one-way ANOVA). (**c**) Golgi staining reveals the dendritic spines from apical dendritic layer of the CA1 region. Scale bar, 2 μm. (**d**) Quantitative analysis of the spine density. (mean ± s.e.m.; $n = 6$ mice per group; $*P < 0.05$, one-way ANOVA). (**e**) The ratio of paired pulses in vehicle- and compound 11-teated control and tau P301S mice (mean ± s.d.; $n = 6$ per group; $*P < 0.05$, one-way ANOVA). (**f**) LTP of fEPSPs (mean ± s.d.; $n = 6$ per group; $*P < 0.05$, one-way ANOVA).

vehicle-treated mice, indicating improved spatial learning (Fig. 6g). On the probe test, the mice treated with compound 11 spent more time in the target quadrant that formerly contained the platform, demonstrating rescue of spatial memory recall by compound 11 (Fig. 6h). The swim speed was not affected by compound 11 (Fig. 6i). These results indicate chronic treatment with compound 11 ameliorates memory deficits in 5XFAD model.

We recently reported that knockout of δ-secretase in 5XFAD mice rescues the synaptic dysfunction such as decreased synaptic density and dendritic spine density, and defective LTP activities[13].

Electron microscopy found that compound 11 notably increased the density of synapse (Fig. 7a,b). Golgi stain found that compound 11 increased the density of spines compared to the vehicle-treated mice (Fig. 7c). Furthermore, compound 11 treatment significantly reversed the LTP deficits in 5XFAD mice, indicating clear restoration of synaptic function by compound 11 (Fig. 7d,e). We also tested the effect of compound 11 on neuroinflammation in 5XFAD and tau P301S mice, and found that compound 11 decreased the density of microglia, and reduced the concentrations of inflammatory cytokines IL-1β and TNFα (Fig. 7f-i).

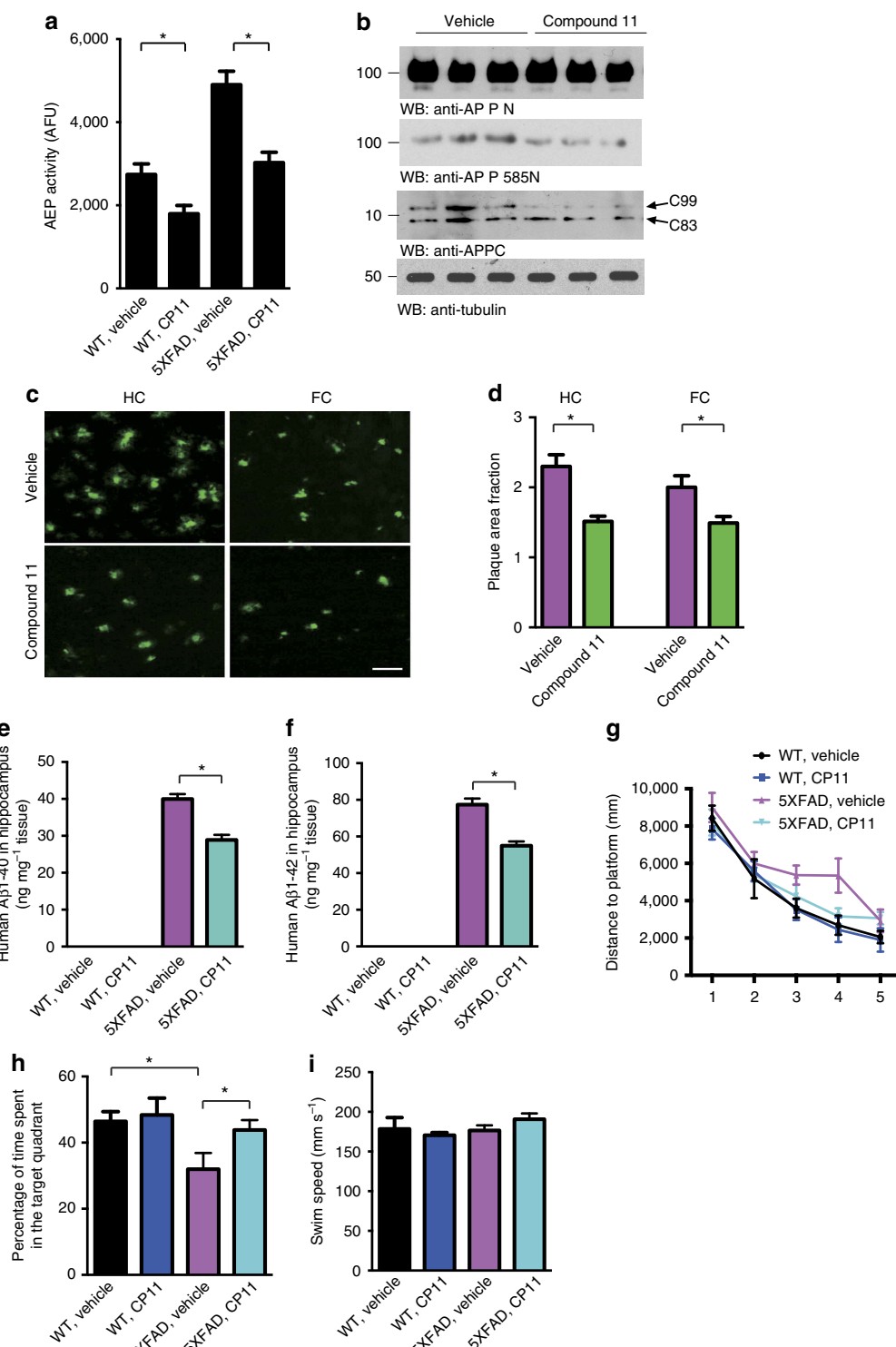

**Figure 6 | Compound 11 inhibits δ-secretase activity, attenuates Aβ deposition and cognitive deficits in 5XFAD mouse model. (a)** δ-secretase activity assay (mean ± s.e.m.; $n = 4$ mice per group; *$P < 0.01$, $t$-test). **(b)** Processing of APP by δ-secretase and BACE1. Compound 11 decreased the δ-secretase-generated APP fragment and BACE1-generated APP fragment (C99), but did not alter the level of full-length APP. **(c)** Thioflavin-S staining of amyloid plaques in the hippocampus, motor cortex and frontal cortex of 5XFAD mouse brain sections. Scale bar, 100 μm. **(d)** Quantitative analysis of amyloid plaques. The density of plaques in 5XFAD mouse brain was decreased by compound 11 (mean ± s.e.m.; $n = 6$ mice per group; *$P < 0.05$, $t$-test). **(e,f)** Aβ1-40 and Aβ1-42 concentration in 5XFAD mice treated with compound 11 or vehicle (mean ± s.e.m.; $n = 6$ mice per group; *$P < 0.05$, $t$-test). **(g)** Morris water maze analysis as distance travelled (mm) for mice treated with compound 11 or vehicle. (mean ± s.e.m.; $n = 10$ mice in wild-type mice vehicle and compound 11 group, $n = 11$ mice in 5XFAD vehicle group, $n = 16$ mice in 5XFAD compound 11 group; *$P < 0.05$, multilevel mixed-effects linear regression model). **(h)** The percentage of time spent in the target quadrant in the probe trail. Mice treated with compound 11 spent more time in the target quadrant than the vehicle-treated mice. (mean ± s.e.m.; $n = 10$ mice in wild-type mice vehicle and compound 11 group, $n = 11$ mice in 5XFAD vehicle group, $n = 16$ mice in 5XFAD compound 11 group; *$P < 0.05$, $t$-test). **(i)** Swim speed of the 5XFAD mice were not altered by treatment with compound 11.

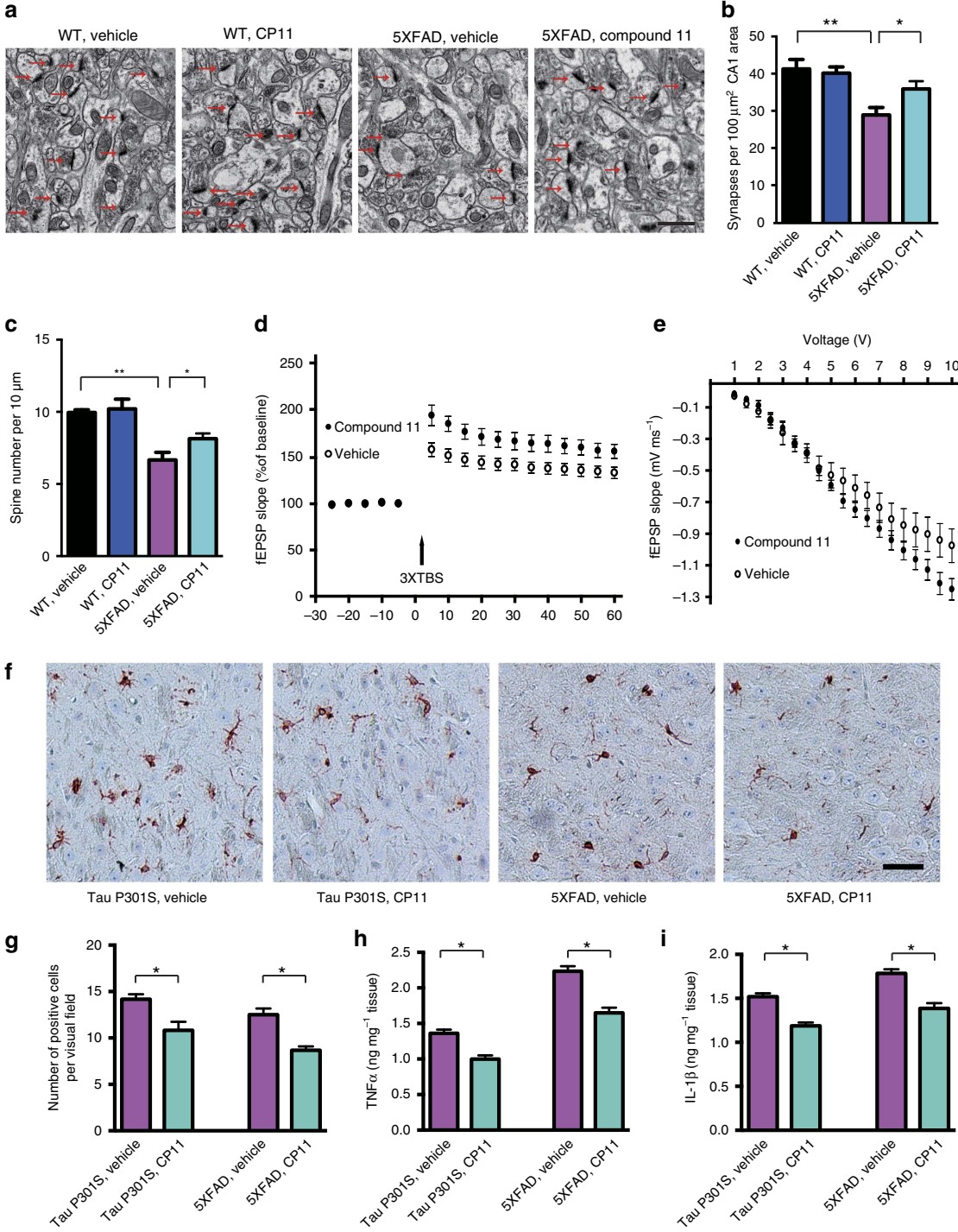

**Figure 7 | Compound 11 ameliorates synaptic loss and restores synaptic dysfunction in 5XFAD mice.** (**a**) Representative electron microscopy of the synaptic structures. Arrows indicate the synapses. Scale bar, 1 μm. (**b**) Quantitative analysis of the synaptic density in vehicle- and compound 11-treated 5XFAD mice (mean ± s.e.m.; $n = 4$ mice per group; *$P < 0.05$, **$P < 0.01$, one-way ANOVA). (**c**) Quantitative analysis of the spine density in Golgi staining (mean ± s.e.m.; $n = 4$ mice per group; *$P < 0.05$, **$P < 0.01$, one-way ANOVA). (**d**) Long-term potentiation (LTP) of fEPSPs was induced by 3 × TBS (4 pulses at 100 Hz, repeated three times with a 200-ms interval). The magnitude of LTP in 5XFAD mice was enhanced by treatment with compound 11 (mean ± s.e.m.; $n = 6$; *$P < 0.05$, one-way ANOVA). (**e**) Synaptic transmission assessed by input/output (I/O) relation between stimuli intensity and fEPSP slope. I/O curves obtained in hippocampal slices prepared from vehicle- and compound 11-treated 5XFAD mice. (**f**) Immunohistochemistry showing the IBA1-positive microglia in vehicle- and compound 11-treated 5XFAD and tau P301S mice. Scale bar, 50 μm. (**g**) Quantitative analysis of IBA1-positive microglia (mean ± s.e.m.; $n = 6$; *$P < 0.05$, t-test). (**h,i**) Concentrations of inflammation mediators IL-1β and TNFα (mean ± s.e.m.; $n = 6$; *$P < 0.05$, t-test).

## Discussion

In the current study, we describe the identification and characterization of small molecule δ-secretase inhibitors with efficacy in the cellular model of acidosis and AD animal models.

A concern in drug discovery projects is the inherent risk of non-specific binding either to the target or off-target. We employed a standard PAINS compound of the benzofurazan family, 11d15 and conducted δ-secretase and GAPDH inhibition

assays. We showed that compound 11 but not 11d15 or GAPDH inhibitor KA selectively antagonized δ-secretase but not GAPDH activity at 2 μM concentration, leading to suppression of OGD-induced neuronal cell death. Remarkably, the dual inhibition mode of this small molecule class explains the biphasic fast-loose and slow-tight mode of δ-secretase. This unique inhibition mechanism provides enormous specificity advantages over the reported peptidic δ-secretase inhibitors that employ reactive warheads to covalently target the active site nucleophile (Cys189). Further, our crystal structure data pave the way for structure-based drug design and optimization of this promising orally bioactive, brain permeable, lead compound.

AD is the most common neurodegenerative disease. It is characterized by the deposition of Aβ and insoluble tau. We recently reported that δ-secretase cleaves both tau and APP in the AD brain. Tau cleavage by δ-secretase generates several fragments that can promote NFT deposition[14]. Furthermore, cleavage of SET by δ-secretase promotes the phosphorylation of tau and neuronal cell death[12,37]. The cleavage of APP by δ-secretase facilitates the processing of APP by BACE1, and promotes the production of Aβ. When δ-secretase was deleted from 5XFAD mice, the Aβ plaque area decreased to about 60% of that in 5XFAD mice, indicating that δ-secretase accounts for about 40% of Aβ production in the brain[13]. All these observations support the notion that inhibition of δ-secretase may rescue the progressive neurodegeneration in AD. We found that chronic administration of compound 11 decreased the N368 truncation and phosphorylation of tau in tau P301S mice by inhibiting the cleavage of tau by δ-secretase. Compound 11 also decreased deposition of Aβ in 5XFAD mice via inhibiting the cleavage of APP by δ-secretase. The restoration of cognitive function by compound 11 indicates that the protective effect of δ-secretase inhibitors *in vitro* may translate into clinical benefit. Synaptic dysfunction is an early and invariant feature of AD, and one of the strongest pathological correlations to dementia in AD (ref. 35). Tau P301S mice and 5XFAD mice show decreased synaptic density and LTP magnitude compared to the non-transgenic mice[34,38]. Compound 11 reversed the synaptic dysfunction as shown by increased synaptic density and LTP magnitude, indicating a 'synaptoprotective' effect of the compound. It is worth noting that the oral bioavailability of compound 11 is ~70% with favourable ADMET profiles. Thus, we have identified a small molecule inhibitor of δ-secretase that is potent, selective, non-toxic and efficacious in a mouse model of AD. Altogether, our data strongly support that compound 11 is a very promising lead compound for optimization via medicinal chemistry and drug development.

## Methods

**Mice and reagents.** Tau P301S mice and 5XFAD mice were from Jackson lab (stock number: 006554 and 008169, respectively), and were bred in a pathogen-free environment in accordance with Emory Medical School guidelines. Only 2-month-old male mice were used in the present study. Sample size was determined by Power and Precision (Biostat). The animals were randomly allocated to experimental groups. Investigators were blinded to the group allocation during the animal experiments. The mice received gavage treatments with vehicle or compound 11 at a dose of 10 mg kg$^{-1}$d$^{-1}$. The protocol was reviewed and approved by the Emory Institutional Animal Care and Use Committee. Rabbit anti-tau N368 (1:1,000), and anti-APP585N (1:500) antibody, which specifically recognize the δ-secretase-derived tau and APP fragments respectively, were described previously[13,14]. GST-HRP (Sigma-Aldrich, GERPN1236, 1:1,000), GFP (Sigma-Aldrich, G1544, 1:1,000), α-tubulin (Sigma-Aldrich, T6074, 1:5,000), HT7 (Thermo, MN1000, 1:1,000), tau5 (Thermo, MA5-12805, 1:1,000), AT8 (Thermo, MN1020, 1:1,000) and AT100 (Thermo, MN1060, 1:1,000), APP N (Calbiochem, MAB348, 1:1,000), and APP C (BioLegend, 802802, 1:5,000). TUNEL *in situ* cell death detection kit was from Roche (Indianapolis, IN). Compound 11 and 11b were purchased from TCI (Portland, OR) and AKOS (Steinen, Germany), and their chemical identity and purity were further confirmed by NMR spectroscopy, mass spectrometry and HPLC (Supplementary Fig. 8). All chemicals not included above

were purchased from Sigma-Aldrich (St. Louis, MO). Recombinant δ-secretase was obtained from Sino Biological (Beijing, China), recombinant cathepsin-S was obtained from Athens Research and Technology (Athens, GA), recombinant caspase-3 and caspase-8 were obtained from Millipore (Billerica, MA). Pala cells were a gift from Dr Colin Watts (University of Dundee, UK), and were cultured in RPMI-1640, supplemented with 10% FBS, 2 mM L-glutamine, 100 units ml$^{-1}$ penicillin, 100 μg ml$^{-1}$ streptomycin.

**High-throughput screening.** Using a micro-high-throughput screening format, an Asinex compound library was screened for potential δ-secretase inhibitors. In 1536-well plates, 625 μg ml$^{-1}$ wild-type mouse kidney lysate was incubated with 16.7 μM library compound and read by an Envision Multilabel plate reader (Ex λ = 360 nm, Em λ = 460 nm) to obtain a background reading. In addition, 1 μM Cbz-Ala-Ala-Asn-AMC was added to initiate the reaction and the fluorescence was measured again after 15 min and the background was subtracted from the final product. The percentage of inhibition, as compared to control wells, was determined and the top 736 compounds were subjected to dose-response confirmatory screening with the kidney lysates from δ-secretase knockout mice and with 50 nM pure active δ-secretase (Sino Biological).

**Enzymatic activity assays.** Cathepsin-S – 100 nM enzyme was pre-incubated with inhibitor in assay buffer (100 mM NaH$_2$PO$_4$ pH 6.5, 100 mM NaCl) for 10 min at 37 °C. The reaction was initiated on addition of 25 μM substrate (Boc-Val-Leu-Lys-AMC (Bachem).

Cathepsin-L – 100 nM enzyme was pre-incubated with inhibitor in assay buffer (100 mM Sodium Acetate pH 5.5, 1 mM EDTA) for 10 min at 37 °C. The reaction was initiated on addition of 25 μM substrate (D-Val-Leu-Lys-AMC (Bachem).

Caspase-3 and 8 – The enzyme was pre-incubated with inhibitor in assay buffer (100 mM HEPES pH 7.4, 0.1% CHAPS, 10% Sucrose) for 10 min at 37 °C. The reaction was initiated on addition of 25 μM substrate (Ac-Asp-Glu-Val-Asp-AMC (Bachem).

δ-secretase – 50 nM enzyme was pre-incubated with inhibitor in assay buffer (50 mM Sodium Citrate pH 5.5, 0.1% CHAPS, 60 mM Na$_2$HPO$_4$, 1 mM EDTA) for 10 min at 37 °C. The reaction was initiated on addition of 10 μM Cbz-Ala-Ala-Asn-AMC (Bachem).

GAPDH activity was assessed using the GAPDH Activity Assay Kit (Biovision) according to the manufacture's instructions.

**IC$_{50}$ assay.** To determine the IC$_{50}$ values of the compounds towards δ-secretase, cathepsin-S, cathepsin-L, caspase-3 and caspase-8, purified recombinant enzyme was incubated in appropriate assay buffers in the presence of increasing concentrations of inhibitors. The formation of fluorescent product was monitored in duplicate reactions and the data were fit to appropriate equations to calculate the IC$_{50}$ values.

To determine the IC$_{50}$ of the compounds in intact Pala cells, we incubated various concentrations of each compound with the Pala cells for 2 h at 37 °C. Cells were collected, washed twice with PBS and lysed in Lysis Buffer (20 mM citric acid, 60 mM disodium hydrogen orthophosphate, 1 mM EDTA, 0.1% CHAPS, pH 5.8). The protein concentrations were normalized via a Bradford assay and lysates were assayed with 5 μM Cbz-Ala-Ala-Asn-AMC. IC$_{50}$ values were determined by fitting the data to the equation: fractional enzymatic activity = $1/(1 + ([I]/IC_{50}))$, in which [I] = inhibitor concentration and IC$_{50}$ = inhibitor concentration that yields half-maximal activity. Data were analysed with GraFit version 5.0.11 software package.

**Comet assay.** Human hepatocellular carcinoma HepG2 cells were used to determine the genotoxicity of the compounds. The Comet Assay (single cell gel electrophoresis) was performed according to the protocol provided in the Trevigen Kit (4250-050-K). Briefly, cells were pre-treated for 24 h with vehicle control or 50 μM compounds. Cells were harvested, embedded in low-melt agarose and submerged in Lysis Buffer for 45 min at 4 °C. After incubation in Alkaline Unwinding Solution (300 mM NaOH, 1 mM EDTA) for 20 min, the cells were subjected to electrophoresis in Alkaline Unwinding Solution at 300 mA for 30 min. Slides were washed with 70% ethanol, dried and stained with SYBR Green for 30 min at room temperature. Nicked DNA was measured as percentage of tail DNA. One hundred comets were scored for each sample[39].

**Micronucleus assays.** HepG2 cells were treated with vehicle or 50 μM compound for 24 h. Cells were washed with PBS, followed by incubation at a ratio of 1:19 in a hypotonic solution (75 mM KCl/0.9% NaCl) for 10 min at 37 °C. Next, the cells were fixed with methanol:glacial acetic acid (3:1) for 15 min at 37 °C, then rinsed and dried. Cells were stained with DAPI (2 μg ml$^{-1}$) for 30 min in the dark at room temperature, rinsed with water, dried and mounted with glycerol. One thousand cells per dish were analysed for each experiment; three independent experiments were performed.

**DTT reversibility assays.** The IC$_{50}$ value (listed in Fig. 2) for each compound was added to the δ-secretase reaction, as described above and allowed to react for

15 min. Simultaneously, 10 mM (final concentration) dithiothreitol (DTT) or L-cysteine was added to each reaction and incubated for an additional 15 min. At the end of the second 15 min incubation time, the fluorogenic activity was measured for each sample. The amount of product formed in the presence of DMSO was used to determine the percentage of δ-secretase activity in the presence of each drug.

**Oxygen-glucose deprivation.** Primary cultured mouse cortical neurons were prepared from E18 embryo and cultured in Neurobasal containing B-27 supplement (Invitrogen) at 37 °C in 5% CO₂/95% air. A half medium was changed to fresh Neurobasal/B27 every 4 days. The neurons were maintained for 13 days *in vitro* (DIV). Compounds were pre-incubated with primary culture neurons for 30 min. The medium was exchanged for glucose-free DMEM and neurons were de-gassed and incubated at 37 °C, 95% N₂/5% CO₂ for 4 h with compounds. the medium was exchanged for DMEM and supplemented with compounds, then the neurons were reperfused for 18 h under normoxic conditions. The neuronal lysates were prepared for δ-secretase assay. Moreover, immunoblotting analysis was conducted with the neuronal lysates.

**Enzyme inhibition assays.** To determine the inhibition constants and the mechanism by which compounds BB1, 11 and 38 inhibit δ-secretase, the steady-state kinetic parameters for the hydrolysis of the peptide substrate, Z-AAN-AMC, were determined in the presence or absence of increasing concentrations of inhibitor. In these assays, specified concentrations of the inhibitor were pre-incubated with substrate for 10 min at 37 °C, then 50 nM δ-secretase was added to initiate the reaction, which was quenched after 10 min. The RFU values of the reaction product were converted to micromolar values with an AMC standard curve and the final reaction rates were plotted against substrate concentration and globally fit to equations representative of competitive inhibition (equation 1), noncompetitive inhibition (equation 2), mixed inhibition (equation 3) and uncompetitive inhibition (equation 4) using a nonlinear least fit squares approach by GraFit version 5.0.11.

$$v = V_{max}[S]/([S] + K_m(1 + [I]/K_{is})) \tag{1}$$

$$v = V_{max}[S]/([S](1 + [I]/K_i) + K_m(1 + [I]/K_i)) \tag{2}$$

$$v = V_{max}[S]/([S](1 + [I]/K_{ii}) + K_m(1 - [I]/K_{is})) \tag{3}$$

$$v = V_{max}[S]/([S](1 + [I]/K_{ii}) + K_m) \tag{4}$$

In the equations, $K_{ii}$ is the intercept $K_i$, and $K_{is}$ is the slope $K_i$. The mode of inhibition was determined by the best fit of the data to equations 1–4. Visual inspection of the fits, and a comparison of the standard errors, was used to confirm these assignments.

**Time course inactivation assays.** Progress curves were generated by incubating 5 μM Z-AAN-AMC and the specified concentration of inhibitor in assay buffer at 37 °C for 10 min. The reaction was initiated by the addition of 50 nM δ-secretase and quenched after 10 min. The concentration of the product was determined from an AMC standard curve and the data were fit by nonlinear regression. Since the curves were nonlinear, they were fit to equation 5, using the GraFit version 5.0.11 software package,

$$[\text{Product}] = v_i(1 - e^{-k_{obs.app}*t})/k_{obs.app} \tag{5}$$

where $v_i$ is the initial velocity, $k_{obs.app}$ is the apparent pseudo-first order rate constant for inactivation and $t$ is time. Equation 6,

$$k_{obs} = ((1 + [S])/K_m)k_{obs.app} \tag{6}$$

was used to correct the apparent pseudo-first-order inactivation rate constants, obtained from this analysis, for substrate concentration and the pseudo-first-order inactivation rate constants, that is, $k_{obs}$, thus obtained, were plotted against the tested inhibitor concentrations. As the data are consistent with a two-step mechanism of inactivation, they were fit to equation 7,

$$k_{obs} = (k_{inact}[I])/(K_I + [I]) \tag{7}$$

using the GraFit version 5.0.11 software, where $K_I$ is the concentration of inactivator that yields half-maximal inactivation, $k_{inact}$ is the maximal rate of inactivation, and [I] is the concentration of inactivator.

**Protein production.** Human wild-type, N263Q and C189S-prolegumain constructs were cloned, expressed, purified and activated as described[40]. Human cystatin E (hCE) full-length cDNA clone IRAUp969C0894D was purchased from Source BioScience (Nottingham, United Kingdom).

Cystatin E was cloned into the pET22b(+) vector (Novagen) utilizing NcoI and XhoI restriction sites. The resulting expression construct harboured an N-terminal signal peptide needed for periplasmic expression in *E.coli* and a C-terminal His₆-tag for purification. Cystatin E was expressed in *E.coli* Bl21(DE3) cells. For large scale expression, 2 l flasks were filled with 600 ml LB medium (Carl Roth, Karlsruhe, Germany), supplemented with 100 μg ml⁻¹ ampicillin and cells

were grown at 37 °C with agitation at 220 rev min⁻¹. After an OD₆₀₀ of 0.8–1.0 was reached, expression was induced by the addition of 1 mM IPTG (isopropyl β-d-1-thiogalactopyranoside) at 25 °C. Following 18–20 h incubation, cells were harvested by centrifugation (4000 r.p.m., 4 °C, 10 min). Cell pellets were resuspended in 20 ml lysis buffer composed of 20 mM Tris pH 7.5 and 300 mM NaCl. Subsequently, cells were lysed by sonication (4 cycles with 30 s pulses at 40 W with 4 min breaks). The lysate was centrifuged twice at 17,500g and 4 °C for 15 min. For purification, the supernatant containing soluble protein was batch-incubated with Ni-NTA Superflow resin (Qiagen, Hilden, Germany) for 20 min at 4 °C. Following a washing step with 100 ml lysis buffer, bound protein was eluted with lysis buffer supplemented with 250 mM imidazole. Elutions were concentrated and subjected to size exclusion chromatography utilizing a SUPERDEX 75 10/300 GL column (GE Healthcare) pre-equilibrated in a buffer composed of 50 mM citric acid pH 5.5 and 50 mM NaCl.

**Crystallization and structure solution.** N263Q-legumain activated at pH 4.0 was crystallized in complex with a covalent Ac-Tyr-Val-Ala-Asp-chloromethylketone inhibitor (Ac-YVAD-cmk, Bachem) as described previously[7,10]. Crystals were soaked with 20 mM ethylmercurphosphate for 72 h, followed by the addition of 5 mM compound 11b. After another 72 h incubation, crystals were harvested and X-ray data were collected at 100 K on beamline ID23-1 (ESRF, Grenoble) equipped with a Pilatus 6M detector to a resolution of 1.9 Å. Approximately 1,000 images were collected at a wavelength of 1.28348 Å at 0.15° oscillation range and 0.037 s exposure time. Similarly crystals of Ac-YVAD-cmk inhibited legumain were soaked with 5 mM compound 11 and harvested after 48 h incubation. X-ray data was collected at 100 K on beamline ID23-2 (ESRF, Grenoble) equipped with a MarCCD detector to a resolution of 2.2 Å. In addition, N263Q-legumain activated at pH 4.0 was crystallized in a condition composed of 22% PEG 3350 and 0.1 M sodium acetate pH 4.5. 0.2 μl protein solution (20 mg ml⁻¹) were mixed with 0.2 μl precipitant at 293 K and crystals appeared after 1–2 months. Similar to the Ac-YVAD-cmk-legumain crystals, apo-legumain crystals were soaked with 5 mM compound 11 and 11b and harvested after 48 h. For cryoprotection, crystals were soaked in 20% glycerol. X-ray data was collected on beamline ID23-2 at 100 K to a resolution of 2.0 (apo-legumain + compound 11b) or 2.1 Å (apo-legumain + compound 11). Data processing was performed utilizing iMOSFLM (ref. 41) and SCALA from the CCP4 program suite[42]. PDB entry code 4awa then served as a model for molecular replacement using PHASER (ref. 43). Iterative cycles of rebuilding in COOT (ref. 44) followed by refinement in phenix.refine[45] and REFMAC (ref. 46) were carried out. CIF-descriptions of ligands were created utilizing JLigand[47].The final structure was analysed using PROCHECK (ref. 48) and MolProbity (ref. 49). Coordinates and structure factors were deposited with the PDB under entry codes 5LU8, 5LU9, 5LUA and 5LUB. Pymol (ref. 50) was used to prepare figures illustrating structures. 2D ligand-protein interaction diagrams were created using LigPlot+ (ref. 51). Electrostatic surface potentials were prepared with APBS (ref. 52) after assigning charges at pH 7.0 using Pdb2pqr (ref. 53). Surface potentials were contoured at ±10 kT/e.

**Interaction studies.** The sam5BLUE biosensor instrument (SAW instruments) was used to test the interaction of different legumain constructs with compound 11. 7-(morpholin-4-yl)-2,1,3-benzoxadiazol-4-amine (compound 11) was covalently coupled to a COOH-dextran sensor chip at 1 mM concentration utilizing carboxylamide coupling. To test the binding of active site inhibited and free legumain to compound 11, wild-type legumain was activated at pH 4.0 and buffer exchanged via a PD-10 column (GE-Healthcare) equilibrated in 20 mM citric acid pH 4.0 and 20 mM NaCl. Subsequently, a 10 μM legumain solution was prepared and supplemented with 80 μM Ac-YVAD-cmk inhibitor. Active site inhibited and free legumain were then buffer exchanged via a NAP-5 column (GE-Healthcare) equilibrated in 20 mM citric acid pH 5.0 and 20 mM NaCl, resulting in 5 μM protein solutions. Additionally, binding of *in trans* activated C189S-prolegumain alone and complexed with cystatin E was tested. C189S-prolegumain was incubated with active wild-type legumain at a 1:50 molar ratio at pH 3.0 for 24 h. Active protease was covalently blocked by the addition of the Ac-YVAD-cmk inhibitor. Subsequently, *in trans* activated C189S-legumain was purified via size exclusion chromatography utilizing a SUPERDEX 75 10/300 GL column (GE Healthcare) equilibrated in a buffer composed of 50 mM citric acid pH 4.0 and 50 mM NaCl. A 10 μM protein solution was prepared and supplemented with 10 μM cystatin E at pH 5.0. Following 30 min incubation complexed and free legumain were buffer exchanged via a NAP-5 column equilibrated in 20 mM citric acid pH 5.0 and 20 mM NaCl, resulting in a final protein concentration of 5 μM. Likewise, 10 μM solutions of C189S-prolegumain and cystatin E only were also buffer exchanged using a NAP-5 column. Interaction studies were performed at 22 °C, with a running buffer composed of 20 mM citric acid pH 5.0 and 20 mM NaCl at a flow rate of 40 μl min⁻¹. Cystatin E alone and papain (5 μM; Merck, Darmstadt) were used as a control to test for unspecific binding to the chip. Data were analysed using the FitMaster Origin-based software.

**Electron microscopy.** Synaptic density was determined by electron microscopy as described previously[54]. After deep anaesthesia, mice were perfused transcardially with 2% glutaraldehyde and 3% paraformaldehyde in PBS. Hippocampal slices

were post-fixed in cold 1% OsO4 for 1 h. Samples were prepared and examined using standard procedures. Ultrathin sections (90 nm) were stained with uranyl acetate and lead acetate and viewed at 100 kV in a JEOL 200CX electron microscope. Four mice were examined in each group. Synapses were identified by the presence of synaptic vesicles and post synaptic densities.

**Mice brain tissue preparation.** After completion of the behavioural test, mice were deeply anaesthetized with pentobarbital and transcardially perfused with saline, and the brains were rapidly removed. One hemisphere was fixed in 4% phosphate-buffered paraformaldehyde, while the other was snap frozen for biochemical analysis.

**Electrophysiological analysis.** Electrophysiological analysis was carried out as described previously[54]. Briefly, vehicle- and compound 11-treated tau P301S mice were anaesthetized with isoflurane, decapitated, and their brains dropped in ice-cold artificial cerebrospinal fluid (a-CSF). The hippocampi were cut into 400-μm thick transverse slices with a vibratome. A 0.1 MΩ tungsten monopolar electrode was used to stimulate the Schaffer collaterals. The field excitatory post synaptic potentials (fEPSPs) were recorded in CA1 stratum radiatum by a glass microelectrode filled with a-CSF with resistance of 3–4 MΩ. The stimulation output (Master-8; AMPI, Jerusalem) was controlled by the trigger function of an EPC9 amplifier (HEKA Elektronik, Lambrecht, Germany). fEPSPs were recorded under current-clamp mode. Data were filtered at 3 kHz and digitized at sampling rates of 20 kHz using Pulse software (HEKA Elektronik). The stimulus intensity (0.1 ms duration, 10–30 μA) was set to evoke 40% of the maximum f-EPSP and the test pulse was applied at a rate of 0.033 Hz. LTP of fEPSPs was induced by three theta-burst-stimulation (TBS; 4 pulses at 100 Hz, repeated three times with a 200-ms interval). Paired-pulse facilitation (PPF) was examined by applying pairs of pulses, which were separated by 20–500 ms intervals. The magnitudes of LTP are expressed as the mean percentage of baseline fEPSP initial slope.

**Western blot analysis.** The cell pellet or mouse hippocampal tissue was lysed in lysis buffer (50 mM Tris, pH 7.4, 40 mM NaCl, 1 mM EDTA, 0.5% Triton X-100, 1.5 mM $Na_3VO_4$, 50 mM NaF, 10 mM sodium pyrophosphate, 10 mM sodium β-glycerophosphate, supplemented with protease inhibitors cocktail), and centrifuged for 15 min at 16,000$g$. The supernatant was boiled in SDS loading buffer. After SDS-PAGE, the samples were transferred to a nitrocellulose membrane. Western blot analysis was performed with the appropriate antibodies. Full-size images are presented in Supplementary Fig. 16.

**Immunohistochemistry.** Free-floating 30-μm-thick serial sections were treated with 0.3% hydrogen peroxide for 10 min, and then incubated with anti-Aβ antibody or anti-IBA1 antibody (1:500) overnight. The signal was developed using Histostain-SP kit (#956543, invitrogen) according to the manufacturer's instructions.

**Morris water maze.** The mice were trained in a round, water-filled tub (52′ diameter) in an environment rich with extra maze cues. An invisible escape platform was located in a fixed spatial location 1 cm below the water surface independent of a subject's start position on a particular trial. In this manner, subjects needed to utilize extra maze cues to determine the platform's location. At the beginning of each trial, the mouse was placed in the water maze with their paws touching the wall from 1 of 4 different starting positions (N, S, E, W). Each subject was given four trials per day for five consecutive days with a 15-min inter-trial interval. The maximum trial length was 60 s and if subjects did not reach the platform in the allotted time, they were manually guided to it. On reaching the invisible escape platform, subjects were left on it for an additional 5 s to allow for survey of the spatial cues in the environment to guide future navigation to the platform. After each trial, subjects were dried and kept in a dry plastic holding cage filled with paper towels to allow the subjects to dry off. The temperature of the water was monitored every hour so that mice were tested in water that was between 22 and 25 °C. Following the 5 days of task acquisition, a probe trial was presented during which the platform was removed. The percentage of time that was spent in the quadrant, which previously contained the escape platform during task acquisition, was measured over 60 s. All trials were analysed for latency, swim path length and swim speed by means of MazeScan (Clever Sys, Inc.).

**AAV vector packaging and stereotaxic injection.** The AAV particles encoding tau 1–368 under the control of synapsin I promoter were prepared by Viral Vector Core at Emory University. Three-month-old wild-type C57BL/6 J mice were anesthetized with phenobarbital (75 mg kg$^{-1}$). Bilateral intracerebral injection of AAV-GFP, AAV-tau 1-368 was performed stereotactically at coordinates posterior 1.94 mm, lateral 1.4 mm, ventral 2.2 mm relative to bregma. Approximately 2 μl of viral suspension containing $2 \times 10^{11}$ vector genome (vg) was injected in to each point using 10-μl glass syringes with a fixed needle. Compound 11 treatment was initiated one week after virus injection.

**Aβ plaque staining.** Amyloid plaques were stained with Thioflavin-S. The deparaffinized and hydrated sections were incubated in 0.25% potassium permanganate solution for 20 min, rinsed in distilled water and incubated in bleaching solution (2% oxalic acid and 1% potassium metabisulfite) for 2 min. After rinsed in distilled water, the sections were transferred to blocking solution (1% sodium hydroxide and 0.9% hydrogen peroxide) for 20 min. The sections were incubated for 5 s in 0.25% acidic acid, then washed in distilled water and stained for 5 min with 0.0125% Thioflavin-S in 50% ethanol. The sections were washed with 50% ethanol and placed in distilled water. Then the sections were covered with glass cover using mounting solution.

**ELISA.** To detect the Aβ deposition in 5XFAD mice, the mice brains were homogenized in 8× mass of 5 M guanidine HCl/50 mM Tris–HCl (pH 8.0), and incubated at room temperature for 3 h. Then the samples were diluted with cold reaction buffer (phosphate buffered saline with 5% BSA and 0.03% Tween 20, supplemented with protease inhibitor cocktail) and centrifuged at 16,000$g$ for 20 min at 4 °C. The supernatant were analysed by human Aβ42 and Aβ40 ELISA kit according to the manufacturer's instructions (KHB3441 and KHB3481, Invitrogen). The Aβ42 and Aβ40 concentrations were determined by comparison with the standard curve. The concentrations of IL-1β and TNFα in the mice brain were detected using mouse IL-1β and TNFα ELISA kit from eBioscience according to the manufacture's instructions.

**Golgi staining.** Mice brains were fixed in 10% formalin for 24 h, and then immersed in 3% potassium bichromate for 3 days in the dark. The solution was changed each day. Then the brains were transferred into 2% silver nitrate solution and incubated for 24 h in the dark. Vibratome sections were cut at 60 μm, air dried for 10 min, dehydrated through 95 and 100% ethanol, cleared in xylene and coverslipped. For measurement of spine density, only spines that emerged perpendicular to the dendritic shaft were counted. Spines that had head diameters more than three times larger than the diameter of their necks and appeared in more than three planes of the images were identified as mushroom spines. Six images were obtained from each animal.

**HPLC experiments.** The purity of compounds 11 and 11b was analysed using an Ultimate 3,000 HPLC instrument (Thermo Fischer Scientific) equipped with an EC Nucleosil C18 column with dimensions 4 × 250 mm (Macherey–Nagel) at 20 °C. As mobile phases 0.06% trifluoroacetic acid in water (A) and 0.05% trifluoroacetic acid in acetonitrile (B) were used. The gradient was 5% B for 5 min and 5–60% B over 35 min at a flow rate of 1 ml min$^{-1}$. The ultraviolet signal was detected at 220 nm.

**NMR spectra.** NMR spectra were measured on a 600 MHz AVANCE III HDX Bruker spectrometer equipped with a $^1H/^{13}C/^{15}N/^{31}P$ quadruple probe. Samples were either measured in DMSO-d$_6$ or methanol-d$_4$ at 298 K. Standard two-dimensional $^{13}C$ heteronuclear single-quantum coherence (HSQC), heteronuclear multiple-bond correlation (HMBC) and $^1H–^1H$ total correlation spectroscopy (TOCSY) spectra were applied to assign the compounds. The two morpholino groups of 11b could be distinguished based on weak negative NOE break-through correlations to protons of the aromatic ring in the 2D TOCSY. Chemical shifts were referenced to tetramethylsilane (TMS), which was added internally for some spectra. Spectra were processed with Topspin 3.2 (Bruker). Two-dimensional spectra were analysed by Sparky (Goddard, T.D. and Kneller, D.G., 2008, Sparky, Version3, University of California, San Francisco, USA).

**Statistical analysis.** Statistical analysis was performed using either Student's $t$-test (two-group comparison) or one-way ANOVA followed by LSD post hoc test (more than two groups), and differences with $P$ values $< 0.05$ were considered significant.

**Data availability.** The authors declare that all data supporting the findings of this study are available within the article and its Supplementary Information files or are available from the corresponding author on request. Crystallographic data have been deposited with the protein database under the entry codes 5LU8, 5LU9, 5LUA and 5LUB.

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

## Acknowledgements

This work is supported by a grant from National Institute of Health (RO1, NS060680) to K.Y., a collaborative grant from the National Natural Science Foundation of China (No. 81528007) to K.Y. and J.-Z.W., a grant from the National Natural Science Foundation of China (No. 81571249) to Z.Z. and the Austrian Science Fund (project P23454-B11). We thank Peter Briza for providing supplementary figure 8.

## Author contributions

K.Y. conceived the project, designed the experiments and wrote the manuscript. O.O., Y.D. and H.F. performed high-throughput screen and characterization of the compounds. E.D., C.C., M.S. and H.B. conducted the crystallization and structure solution, interaction studies, NMR spectra and HPLC experiments. Z.Z., X.L., S.S.K., M.S., S.-P.Y., X.L. and J.-Z.W. conducted the other experiments, assisted with data analysis and interpretation and critically read the manuscript.
