## [Peer Review File · Nature Communications]

Reviewers' comments:

Reviewer #1 (Remarks to the Author):

Zhang et al., presented a very interesting work regarding "Inhibition of delta-secretase improves cognitive functions in mouse model of Alzheimer's disease". Asparagine endopeptidase (AEP) acts as a delta-secretase to cleave both APP and tau, which regulate Abeta and tau pathology in Alzheimer's disease (AD). The authors claimed that an orally bioactive and brain permeable delta-secretase inhibitor, term compound 11, is identified for AD treatment. Moreover, the authors found that a dual active-directed and allosteric inhibition made of this compound family. Lastly, the authors also have found that chronic treatment of such compound 11 on tau P301S and 5XFAD transgenic mice can reduce tau and APP cleavage, decreases synapse loss and increases LTP. However, there are several questions need to be addressed rigorously before this manuscript can be published:

1. Since the authors show binding of compound 11 to delta-secretase in Figure 2. The authors need to demonstrate, if there is any data, the crystallographic data, i.e. space group, cell constant, resolution, Unique reflections, RMSD bonds, and RMSD angles, etc. It may be better to show co-crystal structure of compound 11 in delta-secretase active site.
2. It is important to show that PK/PD of compound 11 in pre-clinical animal models. For example, it is necessary to demonstrate that compound 11 time- and dose-concentrations-dependent inhibition of A β generation in plasma and brain of 5XFAD mice.
3. Since this compound 11 is a dual active-directed and allosteric inhibitor, it would be very strong if the authors can show pharmacologic effects in vivo of oral administration of compound 11 in 5XFAD mice in a time- and dose-dependent manner although the authors show only one dose at one time point (Fig. 4a).
4. Because of high expression of delta in kidney and B lymphoblastoid Pala cells, the cytotoxicity assays should be done not only in primary neurons but also in kidney cells or B lymphoblastoid Pala cells sensitive individual cells.
5. A general question is raised: The authors claimed that inhibition of delta-secretase decreases AD pathology. However, the authors used 5XFAD, it is more like beta-secretase related. How do we know the compound 11 hits delta-secretase in vivo, not other potential targets as the authors discussed off-targets?

Based on the above questions, the methodology is no problem, solid, statistical analyses seem OK. The findings are interesting and intriguing, but the conclusion draw from this manuscript seems to be a little bit ambitious. A major revision is suggested and several key experiments are recommended.

Reviewer #2 (Remarks to the Author):

This manuscript is well written and describes some interesting results. However, the structure of compound 11 is unique enough (CAS# 842964-18-5; 3 patents and one manuscript; Sexton, J. Z.; Danshina, P. V.; Lamson, D. R.; Hughes, M.; House, A. J.; Yeh, L.-A.; O'Brien, D. A.; Williams, K. P. Development and implementation of a high throughput screen for the human sperm-specific isoform of glyceraldehyde-3-phosphate dehydrogenase (GAPDHS). *Curr. Chem. Genomics* 2011, 5, 30-41.) and close enough to PAINS related compounds that the authors should present further data to help the reader gauge its utility as a probe or lead compound. The authors do a good job of comparing the activity of compound 11 versus delta secretase and GAPDH. All of the other compounds in Figure 1a are interference compounds that show the same activity versus AEP. This may suggest that compound 11 is also an interference compound acting by one of the known routes of compound interference. See: Baell, J. B.; Holloway, G. A. New Substructure Filters for Removal of Pan Assay Interference Compounds (PAINS) from Screening Libraries and for Their Exclusion in Bioassays. *J. Med. Chem.* 2010, 53, 2719-2740 and Thorne, N.; Auld, D. S.; Inglese, J. Apparent activity in high-throughput screening: origins of compound-dependent assay interference. *Curr. Opin. Chem. Biol.* 2010, 14, 315-324.

There are four challenges to be overcome before this manuscript is suitable for publication in Nature Communications.

Potential for fluorescence interference: Compound 11 (MLS000764893 [CID: 1095027]) has been reported as an active in several other assays listed in PubChem. A discussion of this is not found in the manuscript. In at least 13 of the 18 assays bioactivity was measured using fluorescent methods. Moreover, in Suzuki, T.; Tsuji, T.; Okubo, T.; Okada, A.; Obana, Y.; Fukushima, T.; Miyashi, T.; Yamashita, Y. Preparation, structure, and amphoteric redox properties of p-phenylenediamine-type dyes fused with a chalcogenadiazole unit. *J. Org. Chem.* 2001, 66, 8954-8960 the lambda max for a similar compound is reported as 476 nm. The fluorescence max for the AMC reporter in the delta secretase assay is 442 nm. It seems that there would be some "fluorescence interference" caused by the overlap of these two spectra. Along these lines, it is not clear how the data in Supp. Fig 2h relates to the enzyme assay. The concentration of AMC would likely be much lower at an earlier time point where the enzyme-activity was measured. Does compound 11 absorb the fluorescence of AMC at concentrations relevant to the enzyme assay? Please see: Simeonov, A.; Jadhav, A.; Thomas, C. J.; Wang, Y.; Huang, R.; Southall, N. T.; Shinn, P.; Smith, J.; Austin, C. P.; Auld, D. S.; Inglese, J. Fluorescence Spectroscopic Profiling of Compound Libraries. *J. Med. Chem.* 2008, 51, 2363-2371.

Lack of convincing SAR: In one of Baell's recent manuscripts (Baell, J. B.; Ferrins, L.; Falk, H.; Nikolakopoulos, G. PAINS: Relevance to Tool Compound Discovery and Fragment-Based Screening. *Aust. J. Chem.* 2013, 66, 1483-1494) he points out the importance of SAR in the validation of HTS hits. First, Supp. Fig 5a needs to be redrawn to show more clearly where the R-groups are connected to the scaffold; the point of attachment should be clearly indicated. Additionally, the structure of 11b in Supp. Fig 5b is probably incorrect and needs to be corrected.

The SAR reported in Supp. Fig 5a is not convincing. The authors state the "amidation of the primary amine on 11 sabotages its inhibitory activity." Contrary to this statement, however, is that it appears that compound 11d5 has the same activity as 11 (See graph at the bottom of Supp. Fig. 5a). Additionally, the SAR in this "series" appears to be very flat given that the activities of 11d-1,2,3,4,5, 9,10,11,12,14 all appear all to be the same. Therefore, there is no real SAR. This may suggest compound interference. That is, there is an overarching interference effect of the compounds that is not related to structure. The fact that 11b appears to be more active than 11 is also surprising and has not been developed enough in the manuscript. Why would 11b be considered in an SAR investigation of 11?

A more convincing SAR would demonstrated by replacing the morpholino group in 11 with other related cyclic amines such as pyrrolidine, piperidine, and substituted or unsubstituted piperazines. Or monoalkylating the free aniline in 11. Of course, these points are moot if the compound is active only via an interference mechanism.

Compound stability: The authors need to show that both compounds 11 and 11b are stable under the conditions of the assays. 11b is a triaminobenzene that may be unstable the acidic buffer used in the assays. This stability should be tested in the presence and absence of cysteamine.

Redox potential: the compounds should be tested for their ability to produce H₂O₂ as described in Johnston, P. A.; Soares, K. M.; Shinde, S. N.; Foster, C. A.; Shun, T. Y.; Takyi, H. K.; Wipf, P.; Lazo, J. S. Development of a 384-Well Colorimetric Assay to Quantify Hydrogen Peroxide Generated by the Redox Cycling of Compounds in the Presence of Reducing Agents. *Assay Drug Dev. Technol.* 2008, 6, 505-518.

Conclusion: Without this data is difficult to judge the other results of the manuscript. Is compound 11 really active via a binding mechanism? Are the crystal structures meaningful given that X-ray results can't be translated to potent binding (i.e. even weakly bound fragments can be detected in crystal structures)? Are the in vivo activities the result of the compound, a decomposition product or an active metabolite?

Reviewer #3 (Remarks to the Author):

This study reports a discovery of compound 11 as a new drug targeting δ -secretase, which inhibits tau and APP cleavage in vitro and in vivo, and protect against neurodegeneration in AD mice models (PS19 and 5XFAD). The study is based on previous work in the same group showing that δ -secretase is abnormally activated in AD brains and mice models, and genetic inhibition of δ -secretase is beneficial. It is a comprehensive work from HTS, hit-to-lead, to in vitro and in vivo validation, which discovered a new compound with promising therapeutic value.

Major points:

1. The logic flow of Figure 3 is not clear, i.e. the use of cellular OGD model is not well justified in the context of the Alzheimer's disease, although the connection was discussed in the Discussion. If OGD is a model of acidosis, it needs to be shown that OGD causes pH reduction in neurons, leading increased AEP activity (Figure 3 a-c), which contributes to increase tau/APP cleavage. Fig. 3g does show that OGD causes cleavage of endogenous tau and APP in primary neurons. At least, correlation analyses should be performed to link OGD, increase AEP activity and increased tau and APP fragments. Additionally, It would be more relevant to use primary neurons with hTau overexpression to examine the effect of CP11 on tau cleavage (Fig. 3d-f).
2. The authors proposed that δ -secretase inhibition reduces tau cleavage, tau phosphorylation AND tau aggregation. However, the Western blot analysis only shows soluble, monomer forms of total tau and its truncated forms. What are the levels of phosphor-tau, tau oligomers and insoluble/aggregated forms of tau?
3. More data supporting the mechanistic link between δ -secretase-induced tau cleavage and increased phosphor-tau and NFT in PS19 model is needed. As discussed in the Introduction and Discussion, SET could be the mediator between δ -secretase and increased p-tau and neuronal death. It would be informative to examine SET activity/localization in primary neuron or PS19 mice brains with and without compound 11 treatment.

Minor points:

1. It would be nice to show whether compound 11 treatment has any beneficial effect on PS19 or 5XFAD mice with δ -secretase knockout, as the authors already have the model published earlier (ref 19,20). These experiments will provide strong evidence of the specificity of the drug and support the proposed mechanism.
2. Line 292-293: "In contrast, compound 11D15 or KA did not affect the cleavage of tau and APP (Fig. 3d-f)". Fig. 3f only shows the lack of effects of 11D15 and KA on tau; the effects on APP were not shown.
3. Fig. 4b: How is cleave hTau and cleaved mTau distinguished in the first panel needs to be clarified.
4. It would be helpful to examine the effects of compound 11 treatment on neuroinflammation in PS19 and 5X FAD mice, as they should have developed increased gliosis by 5-6 months of age.
5. Fig. 4e and Fig. 6g: The authors used integrated latency/distance (AUC) to compare the learning curve. However, multilevel mixed-effects linear regression model should be used to perform the statistics of the learning curve.

Reviewer #1:

Zhang et al., presented a very interesting work regarding "Inhibition of delta-secretase improves cognitive functions in mouse model of Alzheimer's disease". Asparagine endopeptidase (AEP) acts as a delta-secretase to cleave both APP and tau, which regulate Abeta and tau pathology in Alzheimer's disease (AD). The authors claimed that an orally bioactive and brain permeable delta-secretase inhibitor, term compound 11, is identified for AD treatment. Moreover, the authors found that a dual active-directed and allosteric inhibition made of this compound family. Lastly, the authors also have found that chronic treatment of such compound 11 on tau P301S and 5XFAD transgenic mice can reduce tau and APP cleavage, decreases synapse loss and increases LTP. However, there are several questions need to be addressed rigorously before this manuscript can be published:

1. Since the authors show binding of compound 11 to delta-secretase in Figure 2. The authors need to demonstrate, if there is any data, the crystallographic data, i.e. space group, cell constant, resolution, Unique reflections, RMSD bonds, and RMSD angles, etc. It may be better to show co-crystal structure of compound 11 in delta-secretase active site.

Response: The requested crystallographic data are presented in Supplementary Table 5 (X-ray data collection and refinement statistics).

2. It is important to show that PK/PD of compound 11 in pre-clinical animal models. For example, it is necessary to demonstrate that compound 11 time- and dose-concentrations-dependent inhibition of A β generation in plasma and brain of 5XFAD mice.

Response: As suggested by the reviewer, we tested the PK/PD of compound 11 in 5XFAD mice. We found that compound 11 time- and dose-dependently decreased A β concentrations in plasma and brain of 5XFAD mice (Supplementary figure 8b-g).

3. Since this compound 11 is a dual active-directed and allosteric inhibitor, it would be very strong if the authors can show pharmacologic effects in vivo of oral administration of compound 11 in 5XFAD mice in a time- and dose-dependent manner although the authors show only one dose at one time point (Fig. 4a).

Response: To test the time- and dose-dependent effect of compound 11 in 5XFAD mice, we treated 5XFAD mice with compound 11 (10 mg/kg/d) for different time (1.5 month and 3 month, respectively). We also treated 5XFAD mice with different doses of compound 11 (2 mg/kg/d, 5 mg/kg/d, 10 mg/kg/d) for 1.5 month, and monitored the effect of compound 11 on A β deposition. We found that compound 11 decreased the deposition of A β in a time- and dose-dependent manner (Supplementary Figure 12a-h).

4. Because of high expression of delta in kidney and B lymphoblastoid Pala cells, the cytotoxicity assays should be done not only in primary neurons but also in kidney cells or B lymphoblastoid Pala cells sensitive individual cells.

Response: As suggested by the reviewer, we tested cytotoxicity of the compounds in human embryonic kidney 293 (HEK293) cells and B lymphoblastoid Pala cells (Supplementary Figure 3c-f).

5. A general question is raised: The authors claimed that inhibition of delta-secretase decreases AD pathology. However, the authors used 5XFAD, it is more like beta-secretase related. How do we know the compound 11 hits delta-secretase in vivo, not other potential targets as the authors discussed off-targets?

Response: To confirm that compound 11 exerts the beneficial effect via inhibiting δ -secretase, we injected δ -secretase-generated tau 1-368 fragment into the hippocampus, and treated the mice with compound 11 or vehicle. We found that compound 11 did not reveal any beneficial effect in the mice overexpressing tau 1-368, indicating that compound 11 displays its effect via inhibiting the cleavage of tau at N368 by δ -secretase (Supplementary Figure 9a-h).

To further support the specificity of compound 11 on δ -secretase, we tested the effect of compound 11 on OGD-treated neurons from 5XFAD, 5XFAD/ δ -secretase $^{-/-}$, tau P301S, or tau P301S/ δ -secretase $^{-/-}$ mice. We found that compound 11 attenuates OGD-induced damage of cultured neurons from 5XFAD mice or tau P301S mice, but does not affect the cell death of neurons from 5XFAD/ δ -secretase $^{-/-}$ mice or tau P301S/ δ -secretase $^{-/-}$ mice. These results indicate compound 11 exerts its effect via inhibiting δ -secretase activity (Supplementary Figure 9i, j).

Based on the above questions, the methodology is no problem, solid, statistical analyses seem OK. The findings are interesting and intriguing, but the conclusion draw from this manuscript seems to be a little bit ambitious. A major revision is suggested and several key experiments are recommended.

Reviewer #2:

This manuscript is well written and describes some interesting results. However, the structure of compound 11 is unique enough (CAS# 842964-18-5; 3 patents and one manuscript; Sexton, J. Z.; Danshina, P. V.; Lamson, D. R.; Hughes, M.; House, A. J.; Yeh, L.-A.; O'Brien, D. A.; Williams, K. P. Development and implementation of a high throughput screen for the human sperm-specific isoform of glyceraldehyde-3-phosphate dehydrogenase (GAPDHS). *Curr. Chem. Genomics* 2011, 5, 30-41.) and close enough to PAINS related compounds that the authors should present further data to help the reader gauge its utility as a probe or lead compound.

The authors do a good job of comparing the activity of compound 11 versus delta secretase and GAPDH. All of the other compounds in Figure 1a are interference compounds that show the same activity versus AEP. This may suggest that compound 11 is also an interference compound acting by one of the known routes of compound interference. See: Baell, J. B.; Holloway, G. A. New Substructure Filters for Removal of Pan Assay Interference Compounds (PAINS) from Screening Libraries and for Their Exclusion in Bioassays. *J. Med. Chem.* 2010, 53, 2719-2740 and Thorne, N.; Auld, D. S.; Inglese, J. Apparent activity in high-throughput screening: origins of compound-dependent assay interference. *Curr. Opin. Chem. Biol.* 2010, 14, 315-324.

There are four challenges to be overcome before this manuscript is suitable for publication in Nature Communications.

Potential for fluorescence interference: Compound 11 (MLS000764893 [CID: 1095027]) has been reported as an active in several other assays listed in PubChem. A discussion of this is not found in the manuscript. In at least 13 of the 18 assays bioactivity was measured using fluorescent methods. Moreover, in Suzuki, T.; Tsuji, T.; Okubo, T.; Okada, A.; Obana, Y.; Fukushima, T.; Miyashi, T.; Yamashita, Y. Preparation, structure, and amphoteric redox properties of p-phenylenediamine-type dyes fused with a chalcogenadiazole unit. *J. Org. Chem.* 2001, 66, 8954-8960 the lambda max for a similar compound is reported as 476 nm. The fluorescence max for the AMC reporter in the delta secretase assay is 442 nm. It seems that there would be some "fluorescence interference" caused by the overlap of these two spectra. Along these lines, it is not clear how the data in Supp. Fig 2h relates to the enzyme assay. The concentration of AMC would likely be much lower at an earlier time point where the enzyme-activity was measured. Does compound 11 absorb the fluorescence of AMC at concentrations relevant to the enzyme assay? Please see: Simeonov, A.; Jadhav, A.; Thomas, C. J.; Wang, Y.; Huang, R.; Southall, N. T.; Shinn, P.; Smith, J.; Austin, C. P.; Auld, D. S.; Inglese, J. Fluorescence Spectroscopic Profiling of Compound Libraries. *J. Med. Chem.* 2008, 51, 2363-2371.

Response: As suggested by the reviewer, we discussed the fact that compound 11 has been reported as active in several other assays (Page 6, line 1-6). We fully appreciate the problem of PAINS (pan assay interfering) compounds, and have provided substantial amount of data to support that our hit does not act as a PAIN. For example, we show that compound 11 selectively blocks AEP but not GAPDH, another reported target in HTS assay (Figure 3). Moreover, we have provided structure-activity-relation studies (Supplementary Fig 5). Most importantly, we provided the co-crystal structure analysis to demonstrate that Compound #11 indeed targets AEP.

To exclude the fluorescence interference, we tested whether compound 11 absorb the fluorescence of AMC at concentrations relevant to the enzyme assay. We found that compound 11 up to 7 μ M (10 times of IC50) did not absorb the fluorescence of AMC

(Supplementary Figure 2h). In alignment with this finding, kinetics assay results of compound #11 are qualitatively mirrored by those of compound #11b, which has very different absorption characteristics, thus ruling out possible fluorescence interference.

Lack of convincing SAR: In one of Baell's recent manuscripts (Baell, J. B.; Ferrins, L.; Falk, H.; Nikolakopoulos, G. PAINS: Relevance to Tool Compound Discovery and Fragment-Based Screening. *Aust. J. Chem.* 2013, 66, 1483-1494) he points out the importance of SAR in the validation of HTS hits. First, Supp. Fig 5a needs to be redrawn to show more clearly where the R-groups are connected to the scaffold; the point of attachment should be clearly indicated. Additionally, the structure of 11b in Supp. Fig 5b is probably incorrect and needs to be corrected.

The SAR reported in Supp. Fig 5a is not convincing. The authors state the "amidation of the primary amine on 11 sabotages its inhibitory activity." Contrary to this statement, however, is that it appears that compound 11d5 has the same activity as 11 (See graph at the bottom of Supp. Fig. 5a). Additionally, the SAR in this "series" appears to be very flat given that the activities of 11d-1,2,3,4,5, 9,10,11,12,14 all appear all to be the same. Therefore, there is no real SAR. This may suggest compound interference. That is, there is an overarching interference effect of the compounds that is not related to structure. The fact that 11b appears to be more active than 11 is also surprising and has not been developed enough in the manuscript. Why would 11b be considered in an SAR investigation of 11?

A more convincing SAR would demonstrated by replacing the morpholino group in 11 with other related cyclic amines such as pyrrolidine, piperidine, and substituted or unsubstituted piperazines. Or monoalkylating the free aniline in 11. Of course, these points are moot if the compound is active only via an interference mechanism.

Response: We redraw Supplementary Figure 5 to show more clearly where the R-groups are connected to the scaffold. We also clearly indicated the point of attachment. Moreover, we have sorted the available compounds based on their structures and highlight how these translate into the orthosteric and allosteric protein binding modes, supplementary figure 5a.

We have not carried out a fully blown, systematic SAR study. Instead, the intention of our focused SAR was to demonstrate that the NH₂ group can be modified without jeopardizing the binding affinity of the compound. The exception to this rule is the nitro substitution in compound 11d15 which can be explained by its electron withdrawing properties that significantly change the electronic structure of the aromatic ring system, supplementary figure 5b,c. Together these activity data demonstrate that the NH₂ group provides a highly attractive site for further tuning and improving the pharmacological properties of our lead compound.

The compound 11b has been developed to demonstrate that the activity of compound 11 is not strictly related to the benzofurazan double ring. Benzofurazan can be thiol-reactive IF it is substituted with electron withdrawing groups. This is not the

case in compound 11 and consistently the benzofurazan is not thiol-reactive, supplementary figure 13, 14. This conclusion has been confirmed by compound 11b.

The full characterization and description of the SAR is on page 9 and in revised Supplementary Figure 5.

Compound stability: The authors need to show that both compounds 11 and 11b are stable under the conditions of the assays. 11b is a triaminobenzene that may be unstable in the acidic buffer used in the assays. This stability should be tested in the presence and absence of cysteamine.

As requested by the reviewer, we have confirmed by NMR spectroscopy that both compounds 11 and 11b are stable in the presence and absence of cysteamine, (supplementary figure 14a,b). Along the same line, we found that compound 11 was unreactive towards 0.5 M β -mercaptoethanol (supplementary figure 15).

Redox potential: the compounds should be tested for their ability to produce H₂O₂ as described in Johnston, P. A.; Soares, K. M.; Shinde, S. N.; Foster, C. A.; Shun, T. Y.; Takyi, H. K.; Wipf, P.; Lazo, J. S. Development of a 384-Well Colorimetric Assay to Quantify Hydrogen Peroxide Generated by the Redox Cycling of Compounds in the Presence of Reducing Agents. *Assay Drug Dev. Technol.* 2008, 6, 505-518.

Response: As suggested by the reviewer, we measured the potential for redox activity of these compounds using surrogate horseradish peroxidase-phenol red (HRP-PR) assay. We found that compound 11 and 11b do not produce H₂O₂ in the presence or absence of DTT (Supplementary Figure 14c,d). Furthermore, compound 11 was unreactive towards 0.5 M β -mercaptoethanol, as shown by NMR spectroscopy (Supplementary Figure 15).

Conclusion: Without this data it is difficult to judge the other results of the manuscript. Is compound 11 really active via a binding mechanism? Are the crystal structures meaningful given that X-ray results can't be translated to potent binding (i.e. even weakly bound fragments can be detected in crystal structures)? Are the in vivo activities the result of the compound, a decomposition product or an active metabolite?

Response: As listed above, we followed the reviewer's suggestions and provided substantial data to show that compound 11 exerts its inhibitory action by a binding mechanism that has been elucidated to atomic detail. The crystal structures are meaningful as they are in full agreement with functional data such as the Biosensor-chip measurements which confirm that specific binding can occur via the active site and the allosteric site (Supplementary figure 7).

The reviewer's concerns may be related to the mixed orthosteric and allosteric inhibition mode. While allosteric inhibition is generally challenging to understand,

there is significant precedence with the structurally related caspases which exploit an analogous allosteric mechanism positioned at the equivalent beta strand 6 (Supplementary Figure 6d).

Finally, we have pointed out that there are no indications for compound decomposition. By contrast, compound 11 proved to be stable even under rather extreme conditions such as 0.5 M β -mercaptoethanol (Supplementary Figures 14-15).

Reviewer #3:

This study reports a discovery of compound 11 as a new drug targeting δ -secretase, which inhibits tau and APP cleavage in vitro and in vivo, and protect against neurodegeneration in AD mice models (PS19 and 5XFAD). The study is based on previous work in the same group showing that δ -secretase is abnormally activated in AD brains and mice models, and genetic inhibition of δ -secretase is beneficial. It is a comprehensive work from HTS, hit-to-lead, to in vitro and in vivo validation, which discovered a new compound with promising therapeutic value.

Major points:

1. The logic flow of Figure 3 is not clear, i.e. the use of cellular OGD model is not well justified in the context of the Alzheimer's disease, although the connection was discussed in the Discussion. If OGD is a model of acidosis, it needs to be shown that OGD causes pH reduction in neurons, leading increased AEP activity (Figure 3 a-c), which contributes to increase tau/APP cleavage. Fig. 3g does show that OGD causes cleavage of endogenous tau and APP in primary neurons. At least, correlation analyses should be performed to link OGD, increase AEP activity and increased tau and APP fragments. Additionally, It would be more relevant to use primary neurons with hTau overexpression to examine the effect of CP11 on tau cleavage (Fig. 3d-f).

Response: In Figure 3, we used OGD treatment to trigger the activation of AEP. As suggested by the reviewer, we tested the pH and AEP activity of the neurons after OGD treatment. We found that OGD induce acidosis, leading to increased AEP activity (Figure 3a). We also used primary neurons with hTau overexpression and validated the effect of compound 11 on tau cleavage (Figure 3i).

2. The authors proposed that δ -secretase inhibition reduces tau cleavage, tau phosphorylation AND tau aggregation. However, the Western blot analysis only shows soluble, monomer forms of total tau and its truncated forms. What are the levels of phosphor-tau, tau oligomers and insoluble/aggregated forms of tau?

Response: As suggested by the reviewer, we added the data of phospho-tau, tau oligomers and aggregated forms of tau (Figure 4b).

3. More data supporting the mechanistic link between δ -secretase-induced tau cleavage and increased phosphor-tau and NFT in PS19 model is needed. As discussed in the Introduction and Discussion, SET could be the mediator between δ -secretase and increased p-tau and neuronal death. It would be informative to examine SET activity/localization in primary neuron or PS19 mice brains with and without compound 11 treatment.

Response: We agree with the reviewer that SET may also contribute to the phosphorylation and aggregation of tau. As suggested by the reviewer, we tested the activity of SET, and found that PP2A activity was increased in mice treated with compound 11, indicating compound 11 attenuates the inhibitory effect of SET on PP2A, this may also contribute to the decreased phosphorylation of tau in compound 11-treated mice (Figure 4d).

Minor points:

1. It would be nice to show whether compound 11 treatment has any beneficial effect on PS19 or 5XFAD mice with δ -secretase knockout, as the authors already have the model published earlier (ref 19,20). These experiments will provide strong evidence of the specificity of the drug and support the proposed mechanism.

Response: To confirm that compound 11 exerts the beneficial effect via inhibiting δ -secretase, we injected δ -secretase-generated tau 1-368 fragment into the hippocampus, and treated the mice with compound 11 or vehicle. We found that compound 11 did not reveal any beneficial effect in the mice overexpressing tau 1-368, indicating that compound 11 displays its effect via inhibiting the cleavage of tau at N368 by δ -secretase (Supplementary figure 9a-h).

To further support the specificity of compound 11 on δ -secretase, we tested the effect of compound 11 on OGD-treated neurons from 5XFAD, 5XFAD/ δ -secretase^{-/-}, tau P301S, or tau P301S/ δ -secretase^{-/-} mice. We found that compound 11 attenuates OGD-induced damage of cultured neurons from 5XFAD mice or tau P301S mice, but does not affect the cell death of neurons from 5XFAD/ δ -secretase^{-/-} mice or tau P301S/ δ -secretase^{-/-} mice. These results indicate that compound 11 exerts its effect via inhibiting δ -secretase activity (Supplementary Figure 9i, j).

2. Line 292-293: "In contrast, compound 11D15 or KA did not affect the cleavage of tau and APP (Fig. 3d-f)". Fig. 3f only shows the lack of effects of 11D15 and KA on tau; the effects on APP were not shown.

Response: We tested the effect of the compounds on APP processing (revised Figure 3g).

3. Fig. 4b: How is cleave hTau and cleaved mTau distinguished in the first panel needs to be clarified.

Response: We detected two major bands using anti-tau N368 antibody, one at about 50 kDa, and the other one at about 37 kDa. The 50 kDa band was also detected using anti-human tau antibody (HT7), indicating this is the cleaved human tau. The other band at 37 kDa that was not recognized by HT7 antibody is the cleaved mouse tau. We explained this in the figure legends of Figure 4b.

4. It would be helpful to examine the effects of compound 11 treatment on neuroinflammation in PS19 and 5X FAD mice, as they should have developed increased gliosis by 5-6 months of age.

Response: As suggested by the reviewer, we tested the effect of compound 11 on neuroinflammation of PS19 and 5XFAD mice. We found that compound 11 attenuated the activation of microglia, and decreased the concentration of inflammation mediators IL-1 β and TNF α (Figure 7f-i).

5. Fig. 4e and Fig. 6g: The authors used integrated latency/distance (AUC) to compare the learning curve. However, multilevel mixed-effects linear regression model should be used to perform the statistics of the learning curve.

Response: As suggested by the reviewer, we used multilevel mixed-effects linear regression model to perform the statistics of the learning curve (revised Fig. 4g and Fig. 6g).

Reviewers' comments:

Reviewer #1 (Remarks to the Author):

The authors have addressed most of previous referees concerns, which are good. After I read the entire manuscript, I only have one general curiosity, the supplement figure 11 shows compound 11 decreased Abeta and plaques significantly, how much or what proportions of delta secretase regulated Abeta production in AD brains or in 5XFAD AD mouse brains since we understand that Beta- and gamma-secretase processed most of part of Abeta production?

Reviewer #2 (Remarks to the Author):

The authors have done an excellent job of replying to the concerns of the reviewers. This manuscript should be published in Nature Communications when the following minor correction is completed.

(1) Supplementary Figure 5 should be renumbered so there is no duplicate numbering. For example, 14 and 15 now occur twice in the table.

I still have concerns that none of the compounds reported in the manuscript appear to have been purified prior to testing. Yes, their purity was measured, but their purification is never described in the manuscript or supporting information that I can find. Minor impurities in compounds that can't be detected by either HPLC or NMR have been known to lead to assay interference, so this is still an open question.

Reviewer #3 (Remarks to the Author):

The revised manuscript adequately addressed this reviewer's comments, and is improved significantly overall. Additional minor comments for consideration are:

- 1) Double-check the thioflavin S-staining image—the size of the plaques appears to be less than 10 μm , which seem quite unusual.
- 2) Consider use the raw measurements instead of "relative to control", for a better assessment of the effects in the context of existing benchmarks.
- 3) It would be very helpful to include results from wildtype controls (NTG); some of the studies only showed transgenic with only vehicle and CP11-treated.

Reviewer #1:

The authors have addressed most of previous referees concerns, which are good. After I read the entire manuscript, I only have one general curiosity, the supplement figure 11 shows compound 11 decreased Abeta and plaques significantly, how much or what proportions of delta secretase regulated Abeta production in AD brains or in 5XFAD AD mouse brains since we understand that Beta- and gamma-secretase processed most of part of Abeta production?

Response: Delta-secretase does not directly mediate the production of A β . However, it cleaves APP after N585 and generates APP 586-695 fragment. This fragment is readily cut by β -secretase to produce C99 fragment, which is further cut by γ -secretase to generate A β . In our previous report (Zhang Z, et al. Nat Commun, 2015, 6:8762), we found that when δ -secretase was deleted from 5XFAD mice, the A β plaque area decreased to about 60% of that in 5XFAD mice, indicating δ -secretase accounts for about 40% of A β production in the brain. We discussed this issue in the revised manuscript (Page 19, line 21-23).

Reviewer #2:

The authors have done an excellent job of replying to the concerns of the reviewers. This manuscript should be published in Nature Communications when the following minor correction is completed.

(1) Supplementary Figure 5 should be renumbered so there is no duplicate numbering. For example, 14 and 15 now occur twice in the table.

Response: We appreciate the reviewer's suggestions and have renumbered the compounds in Supplementary Figure 5.

(2) I still have concerns that none of the compounds reported in the manuscript appear to have been purified prior to testing. Yes, their purity was measured, but their purification is never described in the manuscript or supporting information that I can find. Minor impurities in compounds that can't be detected by either HPLC or NMR have been known to lead to assay interference, so this is still an open question.

Compounds 11 and 11b were custom-synthesized by AKoS (Steinen, Germany) and TCI (Portland, OR). For this reason we do not report the synthetic protocols. However, we provide the chemical identity of both compounds. Thereby, we are in full accordance to the guidelines of all peer-reviewed journals. For example, the *Nature Chem. Biol.* Guidelines state: "*Authors should provide a statement confirming the source, identity and purity of known compounds that are central to the scientific study, even if they are purchased or resynthesized using published methods. Chemical identity for organic and organometallic compounds should be established through spectroscopic analysis. Standard peak listings for ^1H NMR and proton-decoupled ^{13}C NMR should be provided for all new compounds. For new materials, authors should also provide mass spectral data to support molecular weight identity*"). In the Supplementary Figure 13, we report exactly that: The ^1H NMR, ^{13}C NMR plus the ^{13}C HSQC spectra and the HPLC profiles of compounds 11 and 11b. Now we also add the mass spectra for both compounds

(Supplementary Figure 13h,i). We added this information to the Materials and Methods section of the manuscript. All spectroscopic and analytical data confirm the identity and purity of the compounds.

Reviewer #3:

The revised manuscript adequately addressed this reviewer's comments, and is improved significantly overall. Additional minor comments for consideration are:

1) Double-check the thioflavin S-staining image—the size of the plaques appears to be less than 10 um, which seem quite unusual. □ □

Response: We corrected the scale bar of the thioflavin S staining images in Figure 6c and supplementary Figure 12a,e.

2) Consider use the raw measurements instead of "relative to control", for a better assessment of the effects in the context of existing benchmarks.

Response: We changed to raw measurements in Figure 4 a, d, Figure 6d-f, and Figure 7g-i.

3) It would be very helpful to include results from wildtype controls (NTG); some of the studies only showed transgenic with only vehicle and CP11-treated.

Response: We included the results from wild-type mice treated with vehicle and CP11, respectively (Revised Figure 6a, e-i, Figure 7a-c).

REVIEWERS' COMMENTS:

Reviewer #2 (Remarks to the Author):

The compounds 11 and 11b have been characterized in full accordance with the current guidelines of Nature Communications. The manuscript should be published once the following minor corrections are made.

Note: Line 169: This should be reworded. "The identified compounds contain unsaturated bonds that could potentially be thiol-reactive pan-assay interference (PAINS) compounds..." "Bonds" can't be "compounds". Unsaturated bonds are not an indication of thiol reactivity. For example, benzene is not thiol-reactive.

The bar graph in Figure 5, panel B appears to be incorrectly labeled on the X-axis. In panel E of the same figure, what assay does the IC50 represent?

The numbering of the compounds is now very confusing. For example, supplementary Figure 4 has compounds 10, 11, and 12. There are other compounds numbered 10, 11, and 12 in supplementary Figure 5. (Compound numbering should be consistent throughout the manuscript.)

It may be informative that the most active compound reported, Compound 11 (analogue 23) is almost certainly thiol-reactive, since it contains the same thiol moiety as found in BB1.

Reviewer #2:

1. Line 169: This should be reworded. "The identified compounds contain unsaturated bonds that could potentially be thiol-reactive pan-assay interference (PAINS) compounds..." "Bonds" can't be "compounds". Unsaturated bonds are not an indication of thiol reactivity. For example, benzene is not thiol-reactive.

Response: We reworded the sentence as follow: The identified compounds contain structural moieties with potential thiol reactivity, similarly to pan-assay interference (PAINS) compounds.

2. The bar graph in Figure 5, panel B appears to be incorrectly labeled on the X-axis. In panel E of the same figure, what assay does the IC50 represent?

Response: We believe that the referee meant "Supplementary Figure 5" but not Figure 5, because in Panel E of the same figure contains IC50, whereas Figure 5 does not contain IC50.

We checked the label of X-axis in Supplementary Figure 5B, and they are correct. In panel E, the IC50 represent the IC50 of the compounds against purified AEP (the same as in Figure 1b). We added this information in the figure legend of Supplementary Figure 5.

3. The numbering of the compounds is now very confusing. For example, supplementary Figure 4 has compounds 10, 11, and 12. There are other compounds numbered 10, 11, and 12 in supplementary Figure 5. (Compound numbering should be consistent throughout the manuscript.)

Response: We appreciate the reviewer's comments and renumbered the compounds in Supplementary Figure 5. All of the synthetic derivatives of compound 11 are now labeled as "11dx", in order to distinguish the other positive hits' identity number.

4. It may be informative that the most active compound reported, Compound 11 (analogue 23) is almost certainly thiol-reactive, since it contains the same thiol moiety as found in BB1.

Response: We agree with the reviewer that the most active Compound 11d23 contains thiol moiety as in compound BB1, and probably it is thiol-reactive. However, the parent compound 11 does not contain a thiol moiety, instead, it contains benzoxadiazole not thiadiazole group in BB1.